# Quantitative properties of a feedback circuit predict frequency-dependent pattern separation

Oliver Braganza[1]*, Daniel Mueller-Komorowska[1,2], Tony Kelly[1], Heinz Beck[1,3]*

[1]Institute for Experimental Epileptology and Cognition Research, University of Bonn, Bonn, Germany; [2]International Max Planck Research School for Brain and Behavior, University of Bonn, Bonn, Germany; [3]Deutsches Zentrum für Neurodegenerative Erkrankungen e.V., Bonn, Germany

**Abstract** Feedback inhibitory motifs are thought to be important for pattern separation across species. How feedback circuits may implement pattern separation of biologically plausible, temporally structured input in mammals is, however, poorly understood. We have quantitatively determined key properties of *net*feedback inhibition in the mouse dentate gyrus, a region critically involved in pattern separation. Feedback inhibition is recruited steeply with a low dynamic range (0% to 4% of active GCs), and with a non-uniform spatial profile. Additionally, net feedback inhibition shows frequency-dependent facilitation, driven by strongly facilitating mossy fiber inputs. Computational analyses show a significant contribution of the feedback circuit to pattern separation of theta modulated inputs, even within individual theta cycles. Moreover, pattern separation was selectively boosted at gamma frequencies, in particular for highly similar inputs. This effect was highly robust, suggesting that frequency-dependent pattern separation is a key feature of the feedback inhibitory microcircuit.

*For correspondence:
oliver.braganza@ukbonn.de (OB);
heinz.beck@ukbonn.de (HB)

Competing interests: The authors declare that no competing interests exist.

## Introduction

Efficiently discriminating similar percepts or experiences is a central capability common to invertebrate and vertebrate species. In general terms, such discrimination can be achieved by decreasing the overlap in representations by neuronal ensembles between input and output patterns, a process termed 'pattern separation' (*Cayco-Gajic and Silver, 2019*; *Marr, 1971*; *McNaughton and Morris, 1987*; *Rolls, 2013*). Numerous studies have proposed cellular and circuit mechanisms that support this computation. For instance, sparse divergent inputs, specialized intrinsic properties and feedforward inhibition are thought to generally contribute (*Cayco-Gajic et al., 2017*; *Cayco-Gajic and Silver, 2019*; *Krueppel et al., 2011*; *Mircheva et al., 2019*). Another common feature of most of these models and experimental studies is a critical role of feedback inhibition (*Cayco-Gajic et al., 2017*; *Rolls, 2013*). Feedback circuits differ from the above mechanisms in that they can i) implement direct competition between active cells through lateral inhibition and ii) integrate information about the actual global activity level in a population allowing efficient normalization (*Braganza and Beck, 2018*; *Wick et al., 2010*; *Wiechert et al., 2010*). Indeed, in the insect olfactory system a critical role of such a circuit has been causally demonstrated (*Lin et al., 2014*; *Papadopoulou et al., 2011*).

In mammals, substantial evidence points toward a role of the hippocampal dentate gyrus (DG) for pattern separation during memory formation and spatial discrimination (*Bakker et al., 2008*; *Berron et al., 2016*; *Gilbert et al., 2001*; *Leal and Yassa, 2018*; *Leutgeb et al., 2007*; *McHugh et al., 2007*; *Neunuebel and Knierim, 2014*; *Stefanelli et al., 2016*; *van Dijk and Fenton, 2018*). The DG is thought to subserve this task by converting different types of inputs to sparse,

**eLife digest** You can probably recall where you left your car this morning without too much trouble. But assuming you use the same busy parking lot every day, can you remember which space you parked in yesterday? Or the day before that? Most people find this difficult not because they cannot remember what happened two or three days ago, but because it requires distinguishing between very similar memories. The car, the parking lot, and the time of day were the same on each occasion. So how do you remember where you parked this morning?

This ability to distinguish between memories of similar events depends on a brain region called the hippocampus. A subregion of the hippocampus called the dentate gyrus generates different patterns of activity in response to events that are similar but distinct. This process is called pattern separation, and it helps ensure that you do not look for your car in yesterday's parking space.

Pattern separation in the dentate gyrus is thought to involve a form of negative feedback called feedback inhibition, a phenomenon where the output of a process acts to limit or stop the same process. To test this idea, Braganza et al. studied feedback inhibition in the dentate gyrus of mice, before building a computer model simulating the inhibition process and supplying the model with two types of realistic input. The first consisted of low-frequency theta brainwaves, which occur, for instance, in the dentate gyrus when animals explore their environment. The second consisted of higher frequency gamma brainwaves, which occur, for example, when animals experience something new.

Testing the model showed that feedback inhibition contributes to pattern separation with both theta and gamma inputs. However, pattern separation is stronger with gamma input. This suggests that high frequency brainwaves in the hippocampus could help animals distinguish new events from old ones by promoting pattern separation.

Various brain disorders, including Alzheimer's disease, schizophrenia and epilepsy, involve changes in the dentate gyrus and altered brain rhythms. The current findings could help reveal how these changes contribute to memory impairments and to a reduced ability to distinguish similar experiences.

non-overlapping activity patterns of granule cells (GCs). However, in contrast to the insect olfactory system, the DG feedback circuit is extremely complex, comprising numerous interconnected inter-neuron types (Supplementary Table 1) (*Bartos et al., 2002*; *Dasgupta and Sikdar, 2015*; *Espinoza et al., 2018*; *Ewell and Jones, 2010*; *Freund and Buzsáki, 1996*; *Geiger et al., 1997*; *Harney and Jones, 2002*; *Hefft and Jonas, 2005*; *Kraushaar and Jonas, 2000*; *Larimer and Strowbridge, 2008*; *Lee et al., 2016*; *Liu et al., 2014*; *Lysetskiy et al., 2005*; *Sambandan et al., 2010*; *Savanthrapadian et al., 2014*; *Sik et al., 1997*; *Yu et al., 2015*; *Yuan et al., 2017*; *Zhang et al., 2009*). For instance, interneurons subserving feedback inhibition are also incorporated into circuits mediating feedforward inhibition (*Ewell and Jones, 2010*; *Hsu et al., 2016*; *Lee et al., 2016*) and disinhibition (*Savanthrapadian et al., 2014*; *Yuan et al., 2017*). This makes it difficult to predict the *net* inhibition arising from GC activity.

We reasoned that to assess if feedback inhibition is indeed suitable for the purpose of pattern separation in the DG, it is necessary to determine how efficiently the activity of sparse GC ensembles recruits *net* inhibition, that is the dynamic range and gain of the feedback inhibitory microcircuit. It is furthermore necessary to quantify the spatial and temporal properties of the elicited inhibition, in order to investigate its impact on biologically plausible, temporally structured input. For instance, the DG shows prominent theta oscillations during exploration and distinctive slow-gamma activity during associative memory encoding (*Hsiao et al., 2016*; *Lasztóczi and Klausberger, 2017*; *Pernía-Andrade and Jonas, 2014*; *Sasaki et al., 2018*; *Trimper et al., 2017*). Importantly, both sparsity and temporal oscillations will critically affect a proposed pattern separation function. For instance, feedback inhibition must by definition occur with a delay, a property frequently abstracted away in computational models (*Myers and Scharfman, 2009*; *Rolls, 2016*), but potentially critical during oscillatory activity.

Here, we combine patch-clamp recordings, multiphoton imaging and optogenetics to provide a first quantitative, empirical description of the *net* input-output function of a feedback inhibitory

microcircuit. This includes the spatiotemporal organization of *net* feedback inhibition elicited by a spatially restricted GC population and the *net* short-term dynamics within the feedback microcircuit. Finally, we integrate our data into a biophysically realistic computational model and probe its ability to perform pattern separation. We find a moderate feedback inhibition mediated pattern separation effect during theta modulated input but a substantial separation, particularly of highly similar inputs, during gamma oscillations.

## Results

### Input-output relation of the feedback inhibitory microcircuit

We reasoned that the ultimately relevant parameter for the putative pattern separation effect of feedback inhibition is the *net* inhibition arriving at GCs. We therefore treated the feedback microcircuit as a black-box striving to relate only its *net* input (fraction of GCs active) to its *net* output (feedback inhibition in GCs). To this end, we antidromically recruited feedback inhibitory circuits, while simultaneously recording GC inhibition and population activity (see schematic in *Figure 1A*). Electrical stimulation reliably evoked graded IPSCs in dentate GCs, that increased with stimulation strength (maximal amplitude of 324.1 ± 99.2 pA, n = 8; *Figure 1B*). Feedback IPSCs were completely blocked by 10 µM GABAzine (to 1.5 ± 0.9%, n = 7 cells, P(df = 6, t = 117.4)<0.001, one-sided t-test), as expected (*Figure 1C*). To ascertain that IPSCs were mediated by synaptically activated interneurons rather than interneurons directly recruited by electrical stimulation, we only included slices where inhibition was successfully blocked by glutamatergic antagonists (25 µM CNQX and 50 µM D-APV, 8 of 21 experiments, *Figure 1C*). We also tested if inhibition of glutamate release from mossy fibers, which can be specifically achieved via mGluR2/3 activation by DCG-IV (*Doherty and Dingledine, 1998*; *Toth et al., 2000*), reduces feedback IPSCs. Indeed, we found that IPSCs were reduced to 16.3 ± 6.1% by 0.5 µM DCG-IV (n = 4 cells, P(df = 3, t = 13.73)<0.001, one-sided t-test, *Figure 1C*).

In order to relate the measured IPSCs to the fraction of GCs activated by a given stimulation strength, we used population $Ca^{2+}$ imaging with multibeam two-photon microscopy (*Figure 1A*, see Materials and methods). After bolus-loading GCs with the $Ca^{2+}$ indicator OGB-1-AM (see Materials and methods), antidromic stimulation caused action potential associated $Ca^{2+}$ elevations in a subset of GCs (*Figure 1D*, transients indicated by *). Before quantifying population activity, we verified the reliable detection of single action potentials under our conditions using simultaneous cell-attached recordings from dentate GCs (*Figure 1E*; *Figure 1—figure supplement 1*). Briefly, cells were differentiated into true responders or non-responders on the basis of cell-attached recordings (*Figure 1E,F*; responders green, non-responders grey). A histogram of the peak ΔF/F of non-responders upon a single stimulus was fitted with a Gaussian (*Figure 1F* right, grey dots, grey bars, n = 33) and the threshold set to the quadruple standard deviation of this fit (0.94% ΔF/F, dashed line in *Figure 1F*). We estimated that this threshold would yield approximately equal numbers of false positives and false negatives (*Figure 1—figure supplement 1F*). We additionally controlled for possible errors through variable dye loading and the overestimation of the active cell-fraction through accidental detection of adjacent active cells (*Figure 1—figure supplement 1G,H*, respectively).

Orientation of hippocampal slices may be a critical feature in determining the extent of feedback connectivity. We therefore systematically assessed the magnitude of feedback activation of GCs using imaging in slices obtained from different dorso-ventral levels of the hippocampus (see inset of *Figure 1G*). We found a clear connectivity maximum within horizontal slices obtained at a distance of ~1750 µm from the temporal pole (*Figure 1G,H*; *Bischofberger et al., 2006*). In these and all following experiments we therefore used exclusively slices obtained at 1400–2100 µm from the temporal pole, where the orientation of hippocampal slices matches the orientation of mossy fibers.

Combining the IPSC recordings with population $Ca^{2+}$ imaging allowed us to probe the input-output relationship of the feedback inhibitory microcircuit. Inhibition was recorded in a GC within or immediately adjacent to the imaging field, and stimulation strength was increased gradually (*Figure 1I*). The IPSC saturated at 300µA stimulation strength, where the mean active cell fraction was 2.2 ± 0.7% and the mean IPSC reached 93.1 ± 3.4% of the maximal IPSC (*Figure 1I,J*, n = 20 for imaging, n = 8 for IPSCs including six slices with both). Plotting the IPSC magnitude vs. the cell

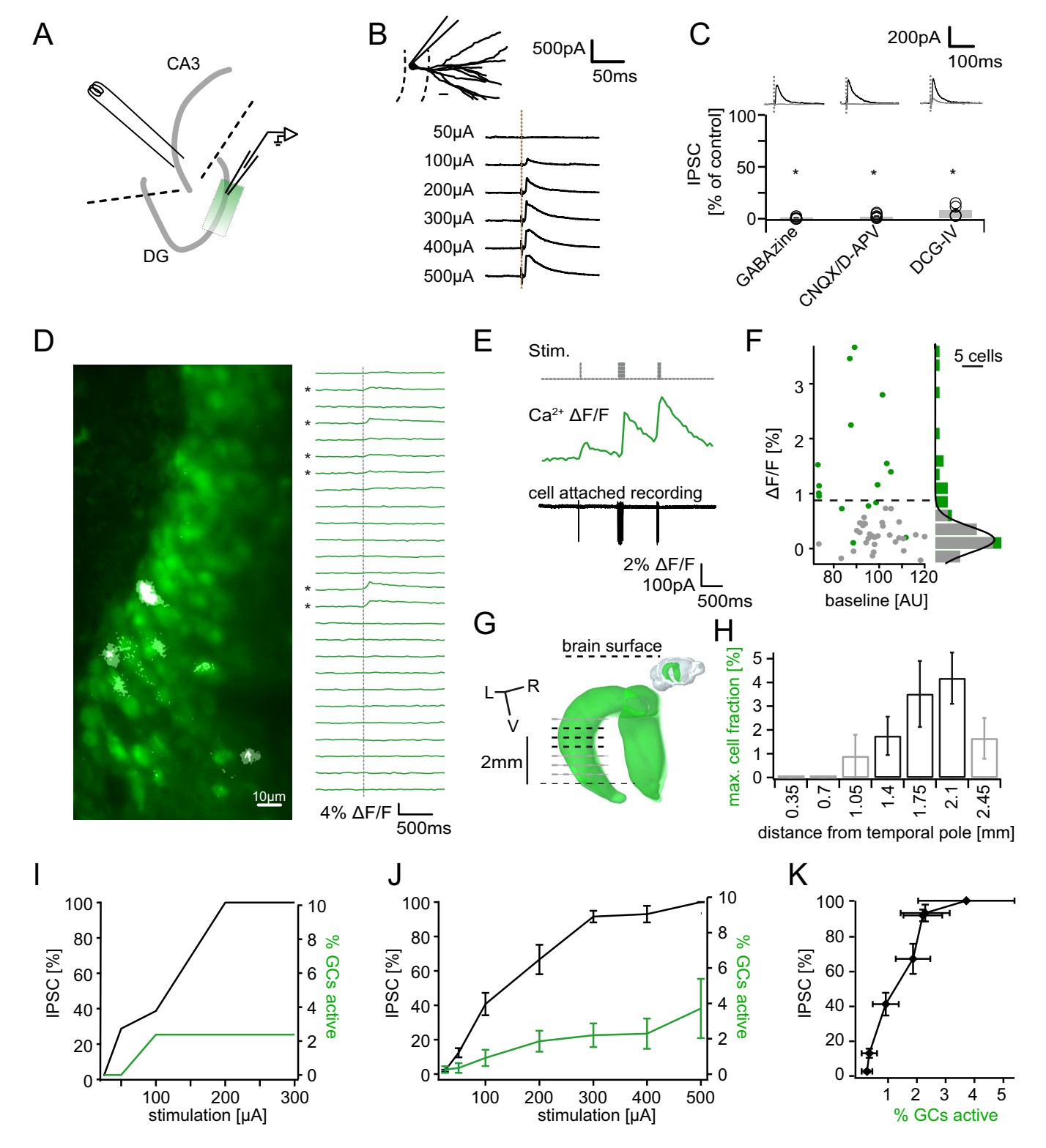

**Figure 1.** Recruitment of feedback inhibition assessed using population Ca$^{2+}$ imaging. Combined population Ca$^{2+}$ imaging and IPSC recordings of GCs during antidromic electrical stimulation. (**A**) Schematic illustration of the experimental setup. Dashed lines represent cuts to sever CA3 backprojections. (**B**) Top: reconstruction of the dendritic tree of a representative GC. Bottom: Feedback IPSC at increasing stimulation strength during stratum lucidum stimulation. (**C**) IPSCs were completely blocked by GABAzine and CNQX + D-APV and largely by DCG-IV. (**D**) Left: overlay of exemplary OGB1-AM-loaded GC population (green) with a ΔF/F map (white). right: traces of ΔF/F over time of a subpopulation of cells depicted on the

*Figure 1 continued on next page*

*Figure 1 continued*

left. (E) Simultaneous cell attached recording and calcium imaging to measure the action potential induced somatic calcium transient amplitude. (F) Scatterplot and histogram of the calcium fluorescence peaks of cells which either did (green) or did not (grey) fire action potentials, as assessed by cell attached recordings. (G) Illustration of the anatomical localization of maximum connectivity plane slices. Short black dashed lines indicate depth at which the slice plane is aligned to the dorsal brain surface. (H) Antidromic stimulation elicited $Ca^{2+}$ transients primarily at this depth (black bars). (I) Normalized IPSC amplitude and activated cell fraction both increase with increasing stimulation strength (example from a single slice). (J) Summary of all slices (K) Summary data plotted to show the increase of inhibition as a function of the active GC fraction.

The online version of this article includes the following figure supplement(s) for figure 1:

**Figure supplement 1.** Detection of single action potential induced calcium transients.

fraction showed that the magnitude of feedback inhibition rises steeply, reaching ~90% with less than 3% of GCs active and complete saturation at $3.7 \pm 1.7\%$ of cells (*Figure 1K*).

## Optogenetic quantification of the recruitment of feedback inhibition

These experiments yielded a first quantitative estimate of the input-output relation of the feedback-inhibitory microcircuit in the DG. We then decided to verify these findings using an alternative method, which allowed spatially controlled and less synchronous GC activation. Mice selectively expressing ChR2$^{(H134R)}$-eYFP in GCs were created by crossing Prox1-Cre mice with Ai32-mice (*Figure 2A*, see Materials and methods). Focal optogenetic stimulation was achieved through a laser coupled into the microscope light path, yielding an 8 µm stimulation spot (*Figure 2B*). Brief (20 ms, 473 nm) light pulses within the molecular layer approximately 40 µm from the dentate GC layer elicited reliable IPSCs in GCs (*Figure 2C*). Increasing the light intensity evoked larger IPSCs that showed clear saturation (*Figure 2C,D*, Power = 7 AU corresponding to 1.7 mW, see Materials and methods). Inhibition was completely blocked by combined application of 40 µM CNQX and 50 µM D-APV (*Figure 2E*, n = 9), confirming that it is recruited via glutamatergic collaterals. The maximal IPSC amplitude obtained optically vs. electrically in experiments in which both stimulations were performed were similar (*Figure 2F*, paired t-test, P(df = 3, t = 1.568)=0.2148, n = 4), indicating that similar maximal inhibition is recruited despite the differences in the activated GC population (distributed vs local; synchronous vs. less synchronous).

In order to relate feedback inhibition to the underlying GC activity levels, we performed systematic cell attached recordings of GCs in the same slices in which inhibition was recorded (~2 cells per slice, *Figure 2—figure supplement 1*). Briefly, we recorded the spatial firing probability distribution in response to focal stimulation for each laser power. We then estimated the mean firing probability of GCs throughout the section, which is equivalent to the expected active GC fraction, by incorporating measurements of the light intensity distribution throughout the slice (*Figure 2G*, black). We additionally estimated an upper and lower bound by assuming either no decay of firing probability with slice depth or isometric decay (*Figure 2G*, grey dashed lines). Combining the input-output relations of IPSCs (*Figure 2D*) and the estimated active cell fraction (*Figure 2G*) again revealed that inhibition is recruited steeply, saturating when approximately 4% of GC are active (*Figure 2H*). Importantly, the resulting recruitment function of inhibition is unlikely to be affected by voltage escape errors (*Figure 2—figure supplement 2*). This is because such errors scale linearly with synaptic conductance and will thus affect the absolute but not the relative amplitude of the somatically measured IPSC. Next, we compared the focal light activation with global activation via a light fiber positioned over the surface of the slice (with powers up to 50 mW, *Figure 2I*). Under global stimulation all cells tested fired APs with 100% reliability and independent of location, even though focal stimulation in direct proximity to the cell led to much lower maximal firing probabilities (*Figure 2I*, middle, $100.0 \pm 0.0$ versus $31.2 \pm 7.1\%$ respectively, paired t-test, P(df = 7, t = 9.74)<0.001, n = 8). At the same time, the maximal IPSC amplitude did not increase further upon global stimulation (*Figure 2I*, right, $356.9 \pm 76.2$ versus $344.3 \pm 77.5$ pA, paired t-test, P(df = 9, t = 1.112)=0.29, n = 10). This implies that additional activation of remote GCs cannot recruit interneurons beyond those activated by local GC populations. Thus, the recruitment of feedback inhibition in the DG is steep, with a dynamic range tuned to sparse populations of GCs (up to 3–4% of cells).

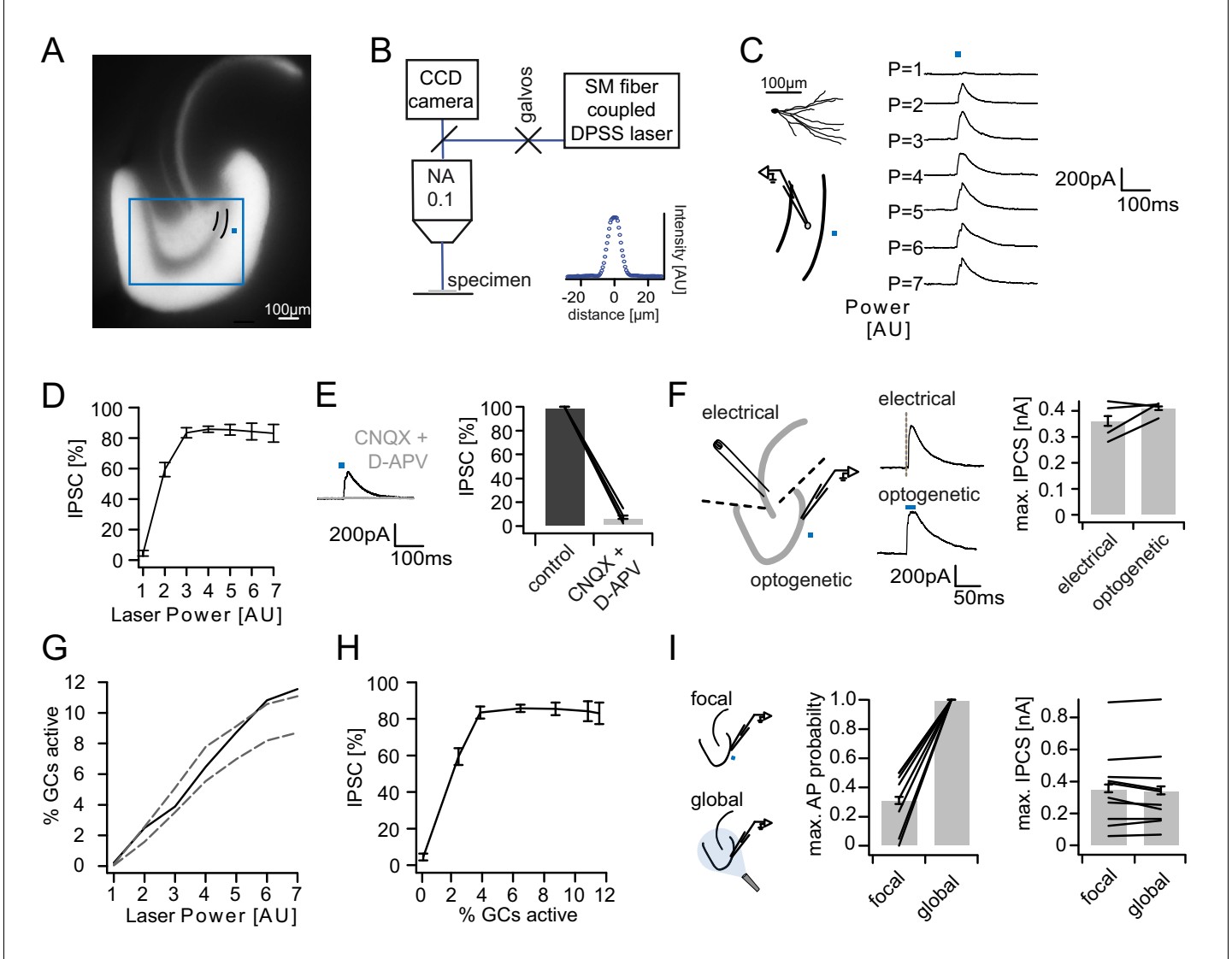

**Figure 2.** Recruitment of feedback inhibition assessed optogenetically. (**A**) EYFP fluorescence in dentate GCs of Prox1/ChR2(H134R)-EYFP transgenic mice. The field of view for rapid focal optogenetic stimulation is indicated by a blue square. A typical stimulation site approx. 40 µm from the GC layer (two short black lines) is indicated by a blue dot. (**B**) Schematic of the microscope setup used to achieve spatially controlled illumination. The inset shows the intensity profile of the laser spot. (**C**) Top left, reconstruction of an Alexa594 filled GC. Left, illustration of optical stimulation. Right, IPSCs following 20 ms light pulses at increasing laser power (p=1 to 7 AU). Each trace represents an average of three trials. (**D**) Summary of IPSC amplitudes from cells in the superior blade (n = 7 cells). IPSC amplitudes were normalized to the maximum amplitude within each cell. (**E**) Optogenetically elicited IPSCs are abolished by glutamatergic blockers (40 µM CNQX + 50 µM D-APV, n = 9). (**F**) Left, Schematic of focal optical and electrical stimulation. Dashed lines indicate cuts to sever CA3 backprojections. Middle, Example traces for IPSCs following electrical or focal optogenetic stimulation. Right, maximal IPSC amplitude for the two stimulation paradigms (361 ± 37 vs. 410 ± 13 pA for electrical and optogenetic stimulation respectively, paired t-test, p=0.28, n = 4) (**G**) The optogenetically activated GC fraction responsible for recruiting the IPSC at the respective laser powers was estimated from systematic cell attached recordings (see *Figure 2—figure supplement 1* for details). The best estimate (black) incorporates measurements of the 3D light intensity profile in the acute slice. Upper and lower bounds were estimated by assuming no firing probability decay with increasing slice depth (upper grey dashed line) or isometric firing probability decay (lower grey dashed line. (**H**) Data from (**D**) and (**H**, best estimate) plotted to show the recruitment of feedback inhibition. (**I**) Comparison of focal optogenetic stimulation to global (light fiber mediated) optogenetic stimulation. Left, Schematic illustration. Middle, Comparison of the AP probability of individual GCs at maximal stimulation power for focal and global stimulation assessed by cell attached recordings. Right, Comparison of the maximal IPSC amplitude under focal and global stimulation for individual GCs. The online version of this article includes the following figure supplement(s) for figure 2:

**Figure supplement 1.** Optogenetically activated cell fraction.

**Figure supplement 2.** Error in somatic IPSC measurements with increasing inhibitory conductance.

**Figure supplement 3.** Absence of single GC induced feedback inhibition.

## Lower limit of feedback recruitment

Previous work has addressed the lower limit of the recruitment of feedback inhibition in various cortical areas (*Jouhanneau et al., 2018*; *Kapfer et al., 2007*; *Miles, 1990*; *Silberberg and Markram, 2007*). The authors report the ability of even a single principal cell to activate feedback inhibitory interneurons and a supralinear increase of inhibition as the second and third principal cells are coactivated (*Kapfer et al., 2007*). Given our findings so far we asked whether single GCs might also suffice to elicit feedback inhibition in the DG. To this end, we performed dual patch clamp recordings and elicited short trains of 10 action potentials at 100 Hz in one cell while monitoring inhibition in the other (*Figure 2—figure supplement 3*, n = 15). However, in contrast to the neocortex (*Kapfer et al., 2007*; *Silberberg and Markram, 2007*) and area CA3 (*Miles, 1990*), we did not find single GC-induced feedback inhibition in any of these experiments, consistent with a recent large scale study reporting that such connections are extremely sparse (0.124%) (*Espinoza et al., 2018*).

## Spatial distribution of feedback inhibition

Recent evidence indicates that inhibition by individual $PV^+$ fast spiking hilar border interneurons is non-uniformly distributed over space, with decreasing connectivity and inhibition at greater distances from the interneuron (*Espinoza et al., 2018*; *Strüber et al., 2015*). To test whether feedback inhibition by the entire ensemble of feedback inhibitory interneurons also displays a spatial gradient, we activated cell populations at 100 µm intervals along the GC layer while recording inhibition in individual GCs (*Figure 3A*). Spatial profiles were recorded for increasing laser powers in cells in the superior as well as inferior blade of the DG (*Figure 3B,C* respectively; n = 8 cells for each blade). IPSC amplitudes across locations and powers were normalized to the maximal IPSC amplitude in each respective cell. This maximal amplitude did not differ between cells in different blades (366 ± 40 vs 390 ± 84 pA for superior and inferior blades, respectively; t-test, P(df = 14, t = 0.258) =0.0686). Next, we investigated the spatial organization of feedback inhibition at stimulation powers at which inhibition had saturated (*Figure 3D,E*). In all GCs tested, the inhibition was greatest when stimulating in the direct vicinity of the recorded cell. Activating cells at increasing distances led to monotonically decreasing IPSC amplitudes for both blades. Importantly, the term distance here refers to the functional distance along the GC layer and not to Euclidean distance. However, inhibition was observed even at the most remote stimulation sites, indicating that even the most remote cells from the contralateral blade can contribute to the activation of feedback inhibition in a given GC. In order to statistically compare the relation of local versus remote inhibition between blades, we defined a remote location in the contralateral blade at 800 µm from the recorded cell (measured along the GC layer and equidistant in all slices; *Figure 3D,E*; grey lines) and compared it to the local IPSC (black lines). Remote inhibition was significantly smaller than local inhibition while no difference between blades or significant interaction was observed (*Figure 3F*; two-way RM ANOVA; Distance: F(1,14)=3.341, p<0.001; Blade: F(1,14)=2.615, p=0.128; Interaction: F(1,14)=3.341, p=0.089). Posttests suggested inhibition of inferior GCs by superior activation might be greater than vice versa. However, the difference was not significant (Sidak's multiple comparison corrected posttest, P (df = 28)=0.932, P(df = 28)=0.051 for local and remote, respectively).

Next, we investigated whether there are differences in the steepness of recruitment of local versus remote inhibition between blades (black and grey, respectively; *Figure 3G,H*). To this end, we calculated the active cell fraction which produces half-maximal inhibition during local or remote stimulation for each individual slice. Comparison of the recruitment between the four groups revealed no differences between blades (*Figure 3I*, two-way RM ANOVA; Distance: F(1,14)=7.889, p=0.014; Blade: F(1,14)=0.5506, p=0.470; Interaction: F(1,14)=0.0976, p=0.759). However, local inhibition was significantly more steeply recruited than remote inhibition (1.99 ± 0.22% vs. 3.17 ± 0.57% active cells for half-maximal inhibition).

Next, we tested if IPSCs elicited by increasing active GC populations differed between local and remote activation with respect to their kinetic properties. Since all previous data showed no indication of blade specific differences the analysis of the kinetics of feedback IPSCs were performed on the pooled data for both blades. Interestingly, local and remote inhibition differed in all tested respects (*Figure 3J–M*, two-way RM ANOVAs with $df_{Distance}$ = 1,183, $df_{cell\ fraction}$ = 6,183 and $df_{interaction}$ = 6,183). Local IPSCs occurred with shorter latency and lower jitter than remote IPSCs (*Figure 3J,K*; Latency: p<0.001, <0.001 and =0.031 for distance, cell fraction and interaction,

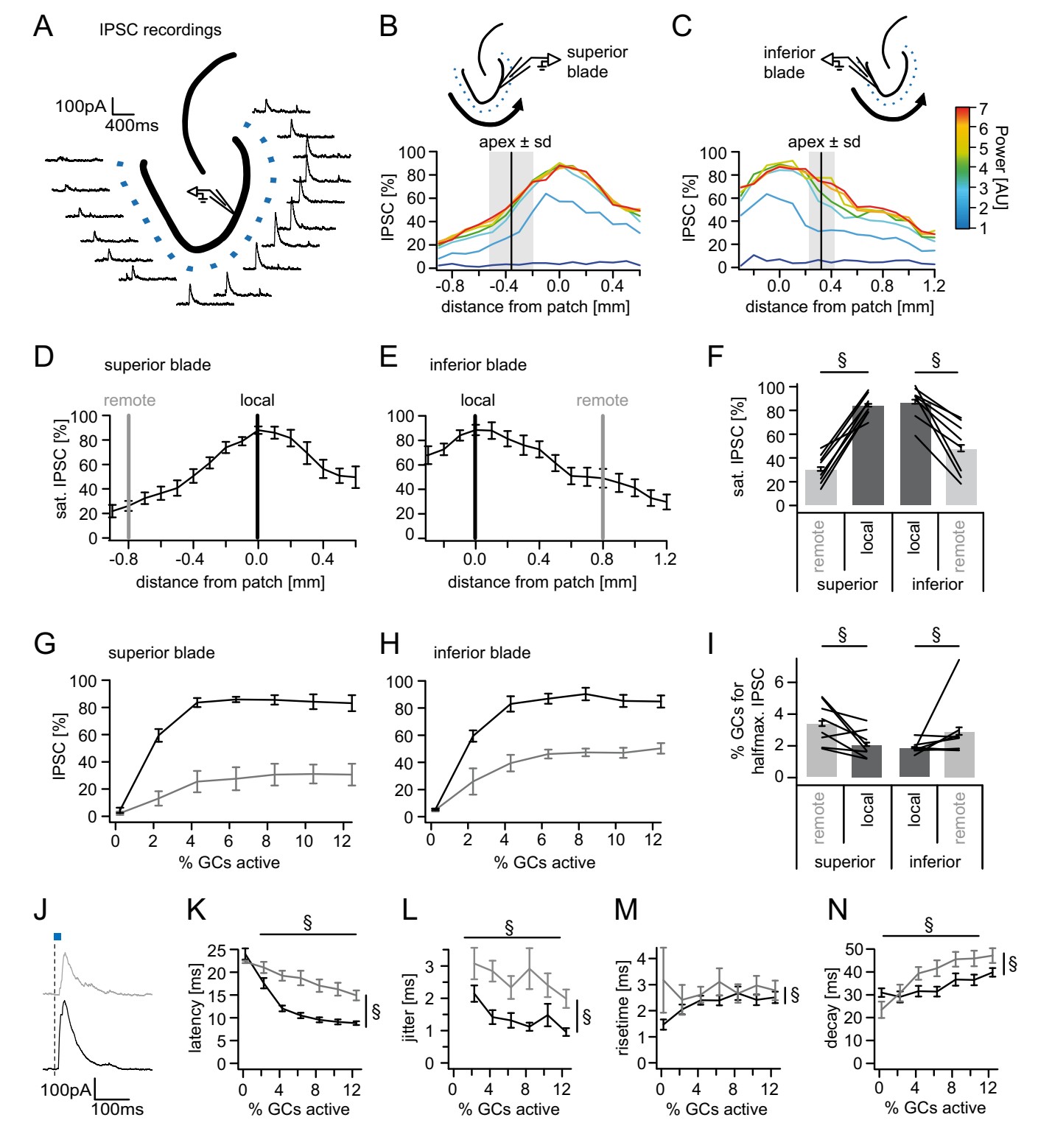

**Figure 3.** Spatial organization of feedback inhibition. Feedback IPSCs recorded from an individual GC while GCs at varying distances were activated. (A) Schematic illustration of the stimulation paradigm and example IPSC traces of an individual trial (p=3). (B, C) Distribution of normalized IPSC amplitudes as a function of laser power and distance from stimulation spot for superior and inferior blade GCs (n = 8 for each blade). The relative location of the DG apex ± standard deviation is indicated by the black bar and grey area respectively. (D, E) IPSC distribution over space at saturation (p≥5). Black and grey bars indicate a local and a remote location at 800 μm from the recorded cell respectively. (F) Comparison of the amplitude of the
*Figure 3 continued on next page*

*Figure 3 continued*

locally and remotely activated IPSCs at saturation (two-way RM ANOVA, overall test significance indicated by §). (G, H) Comparison of the recruitment curves during local (black) or remote (grey) stimulation for superior and inferior blade respectively. (I) Comparison of the cell fraction required for halfmaximal IPSC activation between stimulation sites and blades (two-way RM ANOVA overall test significance indicated by §). (J–M) Temporal properties of IPSCs between local (black) and remote (grey) stimulation. To test for systematic variations of kinetic parameters with increasing active cell fractions as well as stimulation site two-way RM ANOVAs with no post tests were performed. Overall significance indicated by §. (K) Latency from beginning of light pulse to IPSC (L) temporal jitter of IPSCs (SD of latency within cells) (M) 20% to 80% rise time (N) IPSC decay time constant.

respectively; Jitter: p<0.001, =0.037 and =0.707 for distance, cell fraction and interaction, respectively). Furthermore, both latency and jitter decreased as larger populations were activated. IPSCs were also significantly slower in remote versus local inhibition. IPSC rise time was slightly shorter in the larger local IPSCs but did not correlate with the active cell fraction (*Figure 3L*: p=0.010, =0.633 and =0.388 for distance, cell fraction and interaction, respectively). Similarly, decay times were significantly shorter in local versus remote inhibition while they progressively increased with increasing stimulation power (*Figure 3M*; p<0.001, <0.001 and =0.124 for distance, cell fraction and interaction, respectively). These data demonstrate that remote inhibition shows greater delay, greater jitter and slower kinetics than local inhibition.

## Short-term dynamics in the feedback inhibitory microcircuit

Different connections within the feedback inhibitory microcircuit have been shown to variably facilitate or depress during trains of activity (*Savanthrapadian et al., 2014*) (Tabular overview provided in *supplementary file 1*). This makes it difficult to predict the net effect on the short-term dynamics of GC feedback inhibition. We therefore characterized the frequency-dependence of net feedback inhibition using antidromic electrical stimulation as described above (*Figure 4A–C*). In marked contrast to the CA1 region of the hippocampus (*Pothmann et al., 2014*), feedback IPSCs showed strong frequency-dependent facilitation (*Figure 4C*, n = 10 cells, one-way RM ANOVA; Frequency: F(2.69, 29.54)=13.99, p<0.001; Wilcoxon signed rank tests for deviation from unity at each frequency with Bonferroni corrected p-values; p>0.99, p=0.004, p=0.002 and p=0.002 for 1, 10, 30 and 50 Hz, respectively). Furthermore, the facilitation indices significantly increased with increasing stimulation frequency (1 Hz: 0.99 ± 0.07; 10 Hz: 1.41 ± 0.11; 30 Hz: 1.83 ± 0.16; 50 Hz: 2.09 ± 0.19; posttest for linear trend: p<0.0001, $R^2$=0.436). We found no evidence for a spatial gradient of net feedback inhibitory short-term dynamics (*Figure 4—figure supplement 1*).

Because this unusual degree of facilitation may be important in allowing sparse activity of GCs to recruit significant inhibition over time, we further examined the underlying circuit mechanisms. Interestingly, dentate interneuron inputs to GCs appear to be generally depressing (*Supplementary file 1*, blue rows), rendering our finding of pronounced facilitation at the circuit level even more striking. We reasoned that a facilitating excitatory synapse driving feedback interneurons could underlie circuit level facilitation. We therefore measured feedback excitation of hilar neurons by stimulating mossy fiber axons as described above (*Figure 4D–L*). Mossy cells and interneurons were classified according to their morpho-functional properties (*Larimer and Strowbridge, 2008*) (*Figure 4D,E,G, H,J,K*). Cell classification was confirmed using unbiased k-means clustering (*Figure 4K*). We found that feedback excitation of hilar cells displayed marked facilitation, which was similar for both INs and MCs (*Figure 4F,I,L*; n = 9, 12 respectively, two-way RM-ANOVA, Frequency: F(3,57)=6.642, p<0.001; Cell type: F(1,19)=0.0075, p=0.932; Interaction: F(3,57)=0.743, p=0.531). Facilitation indices of hilar cells significantly deviated from one for all frequencies tested (*Figure 4E,F*; n = 23 cells; Wilcoxon signed rank tests with Bonferroni corrected p-values; p<0.001 for all frequencies). These data demonstrate a pronounced frequency-dependent net facilitation of the feedback inhibitory microcircuit, which is supported by strongly facilitating mossy fiber inputs to hilar cells.

## Quantitative properties of the feedback circuit predict frequency-dependent pattern separation

Together, these data indicate that the dentate feedback circuit is able to deliver strong, spatially graded inhibition with a high gain and the ability for temporal integration. To probe how these quantitative properties of the feedback circuit affect the pattern separation capability of the DG, we incorporated them into a biophysically realistic model of the lamellar microcircuit (*Figure 5*) based

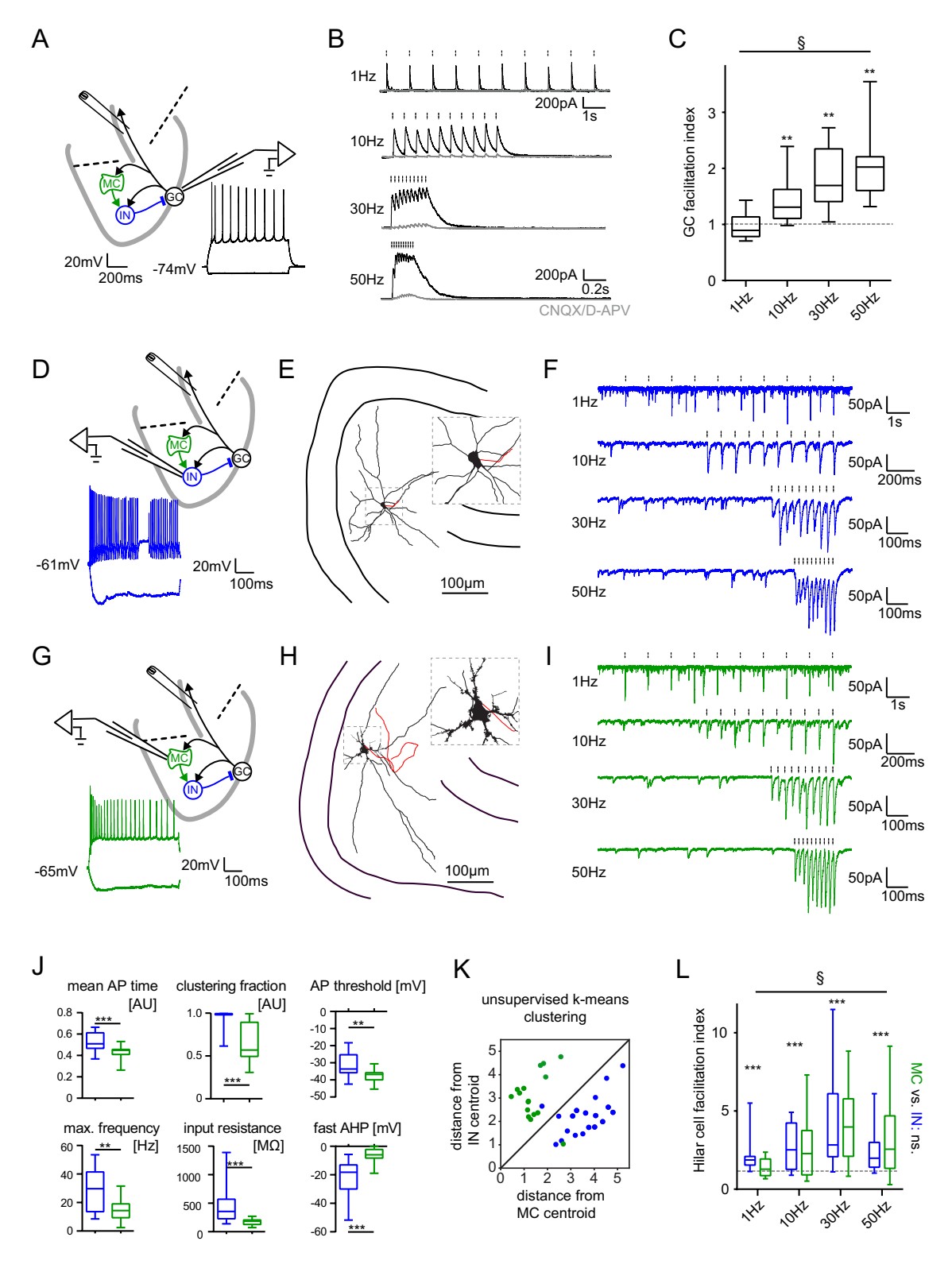

**Figure 4.** Short-term dynamics in the feedback inhibitory microcircuit. Trains of ten antidromic electrical stimulations at 1, 10, 30 or 50 Hz were applied to elicit disynaptic feedback inhibition or excitation of hilar cells (electrical stimulation artifacts were removed in all traces). (**A, D, G**) Schematic illustration of the experimental setup and example traces of voltage responses to positive and negative current injections of GC and hilar cells (dashed lines indicate cuts to sever CA3 backprojections). (**B**) Exemplary GC feedback IPSCs before (black) and after (grey) glutamatergic block (n = 7). (**C**)
*Figure 4 continued on next page*

*Figure 4 continued*

Facilitation indices (mean of the last three IPSCs normalized to the first; n = 10 cells). (D-L) Hilar cells were manually classified into putative interneurons (blue) or mossy cells (green) based on their morpho-functional properties. (E) Reconstruction of biocytin filled hilar interneuron (axon in red). (F) Interneuron EPSCs in response to stimulation trains. (H) Reconstruction of biocytin filled mossy cell (axon in red). (I) Mossy cell EPSCs in response to stimulation trains. (J) Quantification of intrinsic properties of hilar cells (see Materials and methods). (K) k-means clustering based on intrinsic properties of hilar cells (coloring according to manual classification). (L) Facilitation indices of classified hilar cells. (§ indicates significance in one-way RM ANOVA, * show significance in Bonferroni corrected Wilcoxon signed rank tests for deviation from 1).

The online version of this article includes the following figure supplement(s) for figure 4:

**Figure supplement 1.** Frequency dependence of feedback inhibition over space.

on *Santhakumar et al. (2005)*; *Yim et al. (2015)*, making use of their carefully experimentally constrained DG cell-types (*Figure 5A*; *Figure 5—figure supplement 1A*). To maximize our models inferential value we clearly separated a tuning phase, in which we constrained the model by our experimental data, and an experimental phase, in which pattern separation was tested without further changes to the model. In the tuning phase, we first scaled up the model four-fold to contain 400 perforant path afferents (PPs), 2000 GCs, 24 basket cells (BCs), 24 hilar perforant path associated cells (HC) and 60 MCs (*Figure 5A,B*). BCs, HCs and MCs comprise the feedback inhibitory circuit and BCs receive direct PP input thereby additionally mediating feedforward inhibition (*Ewell and Jones, 2010*). We then adapted the spatial extent of the target pools of BC and HC outputs to produce local and global inhibition, respectively, reproducing the experimentally determined spatial tuning of *net* feedback inhibition (*Figure 5C*). We further adjusted synaptic decay time constants and weights in order to reproduce the measured PSCs of hilar neurons and GCs and the empirical recruitment curves (*Figure 5D*, *Figure 5—figure supplement 1*). Finally, we incorporated

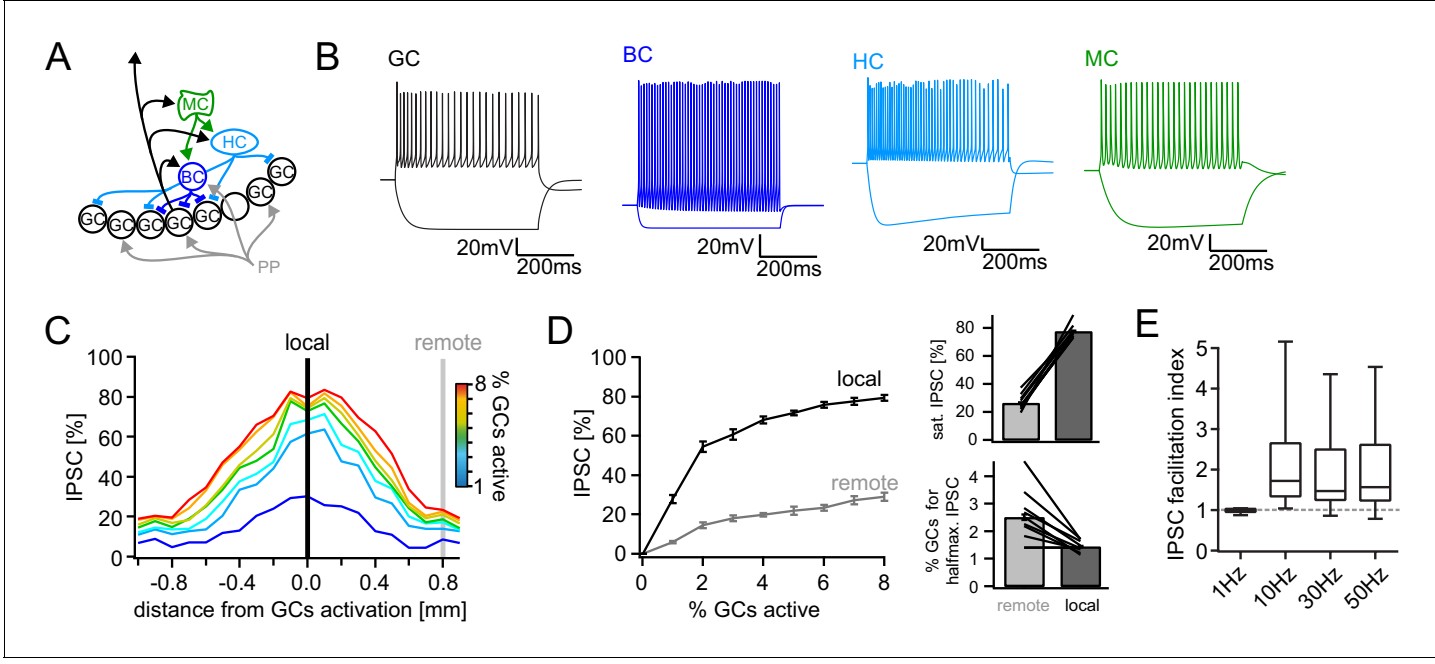

**Figure 5.** Computational model of the DG feedback circuit. A biophysically realistic model of DG was tuned to capture the key quantitative features of the feedback circuit. All analyses were performed as for the real data (including IPSC normalization to maximal IPSC over space and power within each respective cell) (A) Schematic of the model circuit. GC: granule cell, BC: basket cell, HC: hilar perforant path associated cell, MC; mossy cell. (B) Intrinsic responses of model cell types to positive and negative current injections. (C) Spatially graded net feedback inhibition following simulated focal GC activation. (D) Local and remote recruitment curves of the feedback inhibitory circuit (left) and the resulting saturated IPSC amplitudes and GC fractions recruiting halfmaximal inhibition (right). (E) Facilitation indices resulting from simulated, 10 pulse, frequency stimulation of GCs as above.

The online version of this article includes the following figure supplement(s) for figure 5:

**Figure supplement 1.** Model tuning and validation.

facilitation of the experimentally determined magnitude into feedback excitatory mossy-fiber outputs, leading to GC IPSC facilitation in the experimentally observed range (*Figure 5E*, *Figure 5—figure supplement 1B*). Together, these minimal adaptations resulted in a model with remarkably similar properties to our experimental findings (*Figure 5C–E*). We therefore concluded the tuning phase of the model and proceeded to an in silico pattern separation experiment without further changes to the model.

To investigate the implications for pattern separation, we probed the ability of this model to separate PP input patterns with behaviorally relevant temporal structure and varying degrees of overlap (*Myers and Scharfman, 2009*; *Yim et al., 2015*). Specifically, we created input trains with constant mean rate, but with either theta (10 Hz) or slow-gamma (30 Hz) modulation (*Figure 5—figure supplement 1C*), which are prominent during exploration and novelty exposure, respectively (*Sasaki et al., 2018*; *Trimper et al., 2017*). To model rapid pattern separation in a behaviorally relevant timescale we chose an input duration of approximately five theta cycles (600 ms, corresponding to the approximate duration of place cell spiking during traversal of its place field). To obtain a range of input similarities, we generated input patterns in which 24 of 400 PP afferents were activated (*Figure 6A*) and compared pairs of such patterns ranging from no overlap (two separate sets of afferents) to complete overlap (identical trains in the same 24 afferents in both patterns). Each model network was run with 25 input patterns leading to a total of 325 comparisons (data points in *Figure 6C*). To quantify pattern separation we compared input correlation ($R_{in}$) to output correlation ($R_{out}$; *Figure 6B*) both measured as Pearson's R between the population rate vectors over the full 600 ms time window (*Leutgeb et al., 2007*; *Wiechert et al., 2010*).

Our full, tuned model reliably decreased the population vector correlations for similar patterns ($0 < R_{in} < 1$) thereby demonstrating robust pattern separation over the whole range of input similarities ($R_{out} < R_{in}$; *Figure 6C*, left). Next, we isolated the contribution of feedback inhibition to pattern separation by rerunning the same input pattern combinations on the network in which mossy fiber outputs to interneurons were removed (*Figure 6C*, middle). As expected this manipulation decreased interneuron activity and GC sparsity (*Figure 6—figure supplement 1C,D*) leading to impaired pattern separation (*Figure 6D*, noFB). Note that removing mossy fiber outputs also eliminates BC activity through cooperative activation of summating feedforward and feedback inputs (*Ewell and Jones, 2010*). Removal of all inhibitory outputs led to a further decrease in pattern separation, demonstrating the effect of additionally removing feedforward inhibition (*Figure 6C*, right). As expected, each of these manipulations increased both the fraction of active GCs and the activity per GC (*Figure 6—figure supplement 1C,D*). In order to quantify the respective pattern separation effects over the full range of input similarity, we computed the bin wise mean $R_{out}$ (*Figure 6C*, $R_{in}$ bin-width: 0.1, dashed line) and measured the area to the identity line (*Figure 6C*, black lines). The resulting mean $\Delta R_{out}$ was calculated for seven separate random networks, each challenged with theta as well as slow-gamma modulated inputs in each of the three conditions. Both the frequency of the input modulation as well as network manipulations significantly affected pattern separation (*Figure 6D*; two-way RM ANOVA with both factors matching, condition: $F_{(2,12)}=145.1$, $p<0.001$; frequency: $F_{(1,6)}=31.48$, $p=0.001$; interaction: $F_{(2,12)}=11.77$, $p=0.002$; n = 7 random network seeds for these and all subsequent analyses). Specifically, both feedback and feedforward inhibition significantly contributed to pattern separation (Sidak's multiple comparison posttest, $P(df = 12, t = 11.33) <0.001$ and $P(df = 12, t = 5.36)<0.001$, respectively). These results are consistent with the standard account, by which any source of inhibition supports pattern separation by decreasing GC activity (*Figure 6—figure supplement 1C,D*). Notably, the effect of inhibition on GC sparseness was more pronounced during gamma than theta modulated activity, translating to improved pattern separation in the sparser gamma regime (*Figure 6D*, *Figure 6—figure supplement 1C,D*). Remarkably, this increased sparsity in the gamma domain was achieved despite the same excitatory drive from perforant path (*Figure 6—figure supplement 1A,B*), and with less interneuron activity (*Figure 6—figure supplement 1C,D*).

Next, we more closely investigated the isolated pattern separation effects of feedback and feedforward inhibition. To this end, we computed the difference in $R_{out}$ between the respective conditions for each individual comparison (i.e. data point in *Figure 6C*). For instance, the individual comparison shown in *Figure 6A*, will lead to a single $R_{out}$ value in the network with MF inputs to interneurons (full model), which is subtracted from the corresponding $R_{out}$ value in the same network without this input (no FB). This procedure isolates the effect of interest ($\Delta R_{out}$) for each individual

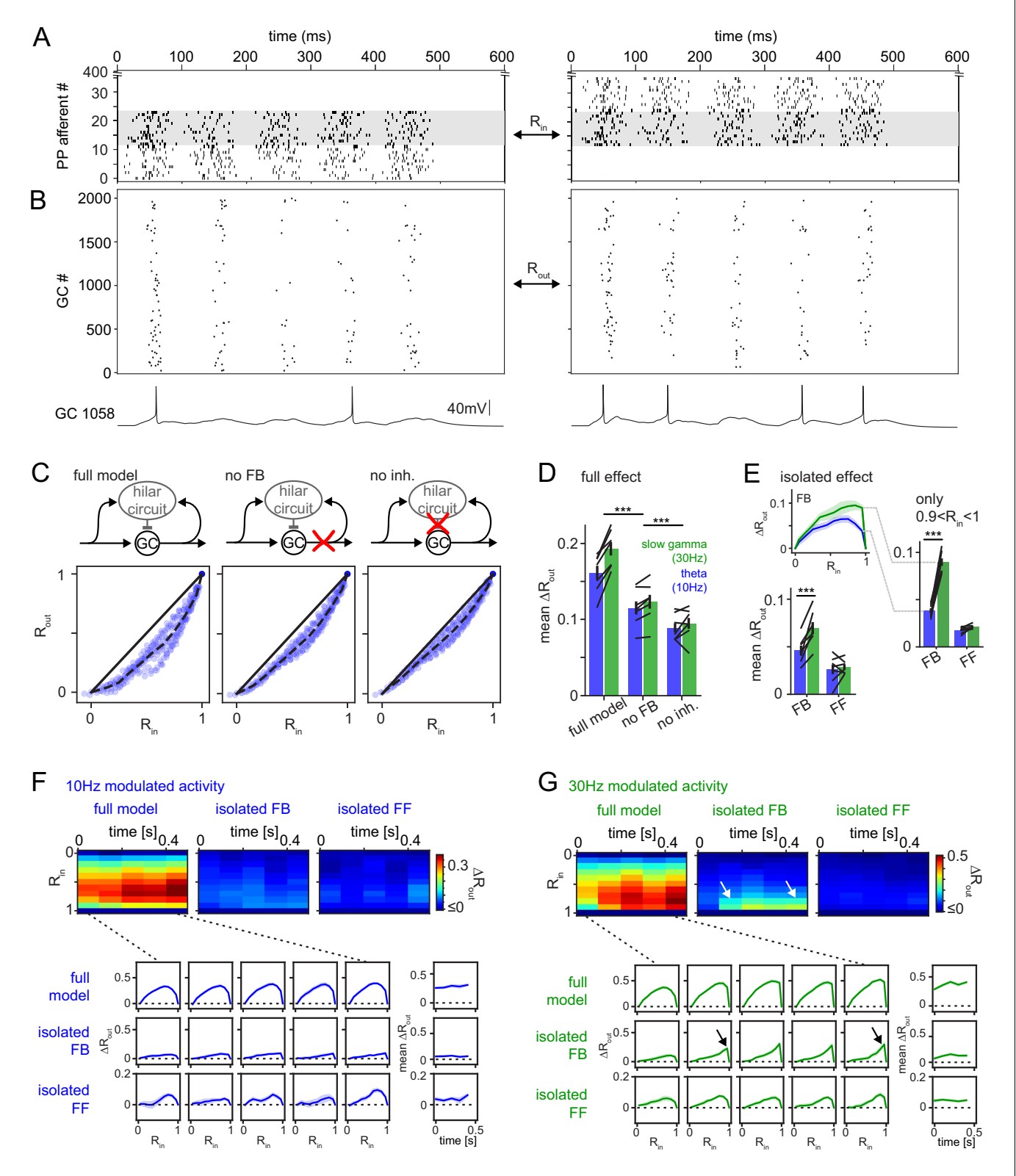

**Figure 6.** Frequency dependent pattern separation of temporally structured inputs. The quantitative DG model was challenged with theta (10 Hz) or slow gamma (30 Hz) modulated input patterns with defined overlap to probe its pattern separation ability. (**A**) Pair of theta modulated perforant path input patterns in which 50% of afferents overlap (grey area). (**B**) Resulting pair of GC output patterns of the full tuned network. Bottom: Representative individual GC underlying the observed patterns. (**C**) Comparison of 325 input pattern pairs and their resulting output pattern pairs. Each pair is

*Figure 6 continued on next page*

*Figure 6 continued*

characterized by its rate vector correlation for inputs ($R_{in}$) and outputs ($R_{out}$), where rates are measured over the full 600 ms time window. Dashed black lines represent the bin-wise mean $R_{out}$ (in $R_{in}$ bins of 0.1). Left: full tuned model, middle: model without mossy fiber inputs to interneurons, right: model without inhibitory synapses. (D) Full pattern separation effects (mean $\Delta R_{out}$) of all three conditions for both frequency domains quantified as the area enclosed by the dashed and unity lines in (C). Black lines represent individual network seeds. Two-way RM ANOVA indicated significance of condition, frequency and interaction, * indicate significance in Sidak's posttests between individual conditions. (E) Isolated effects of feedback and feedforward motifs obtained by pairwise subtraction of $R_{out}$ between conditions for each individual comparison. The inset shows the resulting $\Delta R_{out}$ for each $R_{in}$ bin. The area under the curve quantifies the mean $\Delta R_{out}$ as in (D). Two-way RM ANOVA indicated significance of condition, frequency and interaction. *** indicate p<0.001 in Sidak's posttest. (F). 100 ms time-resolved pattern separation effects of the full model, isolated FB or FF inhibition for theta modulated input (10 Hz). All analyses were performed as above but with rate vector correlations computed for 100 ms time windows. The bottom insets show $\Delta R_{out}$ as a function of input similarity for each time window. The bottom right insets show the evolution of the mean $\Delta R_{out}$ over time. (G) Same as (F) but for slow gamma (30 Hz) modulated inputs. Arrow indicate the region of selectively increased pattern separation. Data in D-G represent mean ± SEM of n = 7 random network seeds.

The online version of this article includes the following figure supplement(s) for figure 6:

**Figure supplement 1.** Activity levels and pattern separation.
**Figure supplement 2.** Robustness over different Similarity Metrics.
**Figure supplement 3.** Isolated pattern separation effects of spatial tuning and MF facilitation.
**Figure supplement 4.** Robustness for shorter analysis time-window.
**Figure supplement 5.** Robustness over various IPSC decay time-constants and over the full gamma range.
**Figure supplement 6.** Robustness for increased feedforward inhibition.
**Figure supplement 7.** Robustness for increased perforant path (PP) drive.

comparison, controlling for other sources of variability. A single pattern separation measure was then obtained as before, as the area under the curve of bin-wise means of these $\Delta R_{out}$ values (*Figure 6E*, bottom). We found a significant effect of both inhibitory motif and frequency domain (*Figure 6E*; two-way RM ANOVA with both factors matching, Motif: F(1,6)=15.58, p=0.008; Frequency: F(1,6)=9.91, p=0.020; Interaction: F(1,6)=76.37, p<0.001). Posttests revealed that the frequency dependence of pattern separation was driven by feedback inhibition (Sidak's multiple comparison posttest: FB: P(df = 6, t = 13.68)<0.001; FF: P(df = 6, t = 1.33)=0.412. Interestingly, this frequency dependence of feedback inhibition mediated pattern separation was particularly pronounced for highly similar input patterns (0.9 < $R_{in}$ < 1; *Figure 6E*, right; Motif: F(1,6)=261.7, p<0.001; Frequency: F(1,6)=108.1, p<0.001; Interaction: F(1,6)=109.5, p<0.001; Sidak's multiple comparison posttest: FB: P(df = 6, t = 15.78)<0.001; FF: P(df = 6, t = 0.98)=0.595). Indeed, feedback inhibitory pattern separation for highly similar input at 30 Hz compared to 10 Hz was more than doubled (from 0.04 ± 0.01 to 0.09 ± 0.01, mean ± SD, Cohen's d = 4.1, *Figure 6E*, right). This again demonstrates feedback inhibitory pattern separation effects beyond those explainable by decreases in GC activity, since comparisons for highly similar inputs are computed on the exact same model runs as comparisons for less similar inputs and thus by definition have the same GC activity levels (also see *Figure 6F,G*, arrows).

It has recently been emphasized, that the assessment of pattern separation can depend critically on the similarity measure used (*Madar et al., 2019*; *Wick et al., 2010*). Therefore, we tested the robustness of this result for two alternative similarity measures, namely normalized dot product (NDP, also known as cosine similarity) and pattern overlap (# of coactive/ # of totally active cells; *Figure 6—figure supplement 2*). The frequency dependence of feedback inhibition-mediated pattern separation, especially for highly similar inputs, proved robust for all three similarity measures.

## Effect of spatial tuning and facilitation of net feedback inhibition

Next, we investigated the specific effects of two interesting empirical findings of the present study, 1) the spatial tuning and 2) the facilitation of the feedback circuit (*Figure 6—figure supplement 3*). To this end, we undertook two targeted, minimal manipulations of the full tuned network. To probe the effect of spatially graded inhibition, we redistributed BC output synapses to a global target pool (the whole GC population), leading to spatially uniform inhibition (global FB; *Figure 6—figure supplement 3B,E*). To probe the effect of facilitation, we removed facilitation from mossy fiber outputs (*Figure 6—figure supplement 3C,E*). We isolated the effects of these manipulations by pairwise comparison to the corresponding full tuned networks as described above (*Figure 6—figure*

supplement 3F–I). The results showed a small but significant contribution of facilitation (~20% of the isolated FB effect for both frequency paradigms), but not spatial tuning to pattern separation (*Figure 6—figure supplement 3G*, left; Wilcoxon signed rank test for deviation from 0, n = 7, Bonferroni corrected p-values: p=0.031 and p=1 respectively for 10 Hz; p=0.031 and p=1 respectively for 30 Hz). We noted that while spatial tuning did not affect mean pattern separation, it appeared to reduce its variability (CoV) for a given input similarity, although the effect was again small (*Figure 6—figure supplement 3G*, right; Wilcoxon signed rank test for deviation from 0, n = 7, Bonferroni corrected p-values: p=0.031 and p=0.750 for tuning and facilitation respectively at 10 Hz; p=0.438 and p>0.999 respectively at 30 Hz).

## Frequency-dependent pattern separation is robust over analysis scales and input strengths

So far, all pattern separation analyses were conducted on the population rate vectors during a 600 ms time window. However, many neural computations are likely to occur on shorter timescales, such as within individual theta (~100 ms) and gamma (~10–33 ms) cycles (*Buzsáki, 2010*; *van Dijk and Fenton, 2018*). Indeed, the time window in which correlation is recorded can nontrivially affect the resulting correlation, depending on the timing of spikes within it (*Madar et al., 2019*). We therefore first computed the networks pattern separation ability within 100 ms time windows, revealing i) the pattern separation ability within such short timescales and ii) the temporal evolution of pattern separation throughout a 600 ms stimulus presentation (*Figure 6F,G*). We find that pattern separation occurs even within a single theta cycle, including a contribution of feedback inhibition in both frequency paradigms (mean $\Delta R_{out} > 0$ within the first 100 ms bin, Wilcoxon signed rank test with Bonferroni corrected p-values: p=0.031,=0.031 for full and FB effect respectively in both paradigms). While mean $\Delta R_{out}$ did not differ between frequency paradigms within this first time window, it was significantly elevated in the 30 Hz paradigm in all subsequent time windows (full model effect, two-way RM ANOVA, p<0.001,<0.001 and=0.004 for time-bin, frequency and interaction respectively, Sidak's posttest p=0.234 for 1st bin and p<0.001 for all subsequent bins). Again, the selective increase during slow-gamma modulated inputs was driven by feedback inhibition (isolated FB effect, two-way RM ANOVA, p=0.007,<0.001 and=0.041 for time-bin, frequency and interaction respectively, Sidak's posttest p=0.708 for 1st bin and p<0.002 for all subsequent bins), including a contribution from MF facilitation (*Figure 6—figure supplement 3*). As above, the effect was predominantly driven by the separation of highly similar input patterns (isolated FB effect, $R_{in} > 0.5$; two-way RM ANOVA on last time-bin, p<0.001,=0.010 and<0.001 for $R_{in}$-bin, frequency and interaction respectively, Sidak's posttest on differences between frequency paradigms for each input similarity: p=1 for $R_{in} < 0.6$ and p=0.032 to p<0.001 for $R_{in}$ = 0.6 to 0.9). These results were robust when analysis time windows were decreased even further (to the duration of a slow gamma cycle, 33 ms, *Figure 6—figure supplement 4*). This 33 ms resolved analysis additionally reveals that the pattern separation effect, particularly of feedback inhibition, ramps up within a 100 ms window, becoming effective only at the end of a theta cycle (*Figure 6—figure supplement 4A*).

Next, we asked if the frequency dependence of feedback inhibitory pattern separation was sensitive to variations of the inhibitory decay time constants and if there might be an interaction between these decay time constants and the frequency range at which pattern separation is most effective (*Figure 6—figure supplement 5*). Remarkably, we found the differential effect between 10 and 30 Hz to be highly robust across a range of different decay time-constants (0.5x to 5x of the experimentally matched decay, *Figure 6—figure supplement 5A–C*, *Supplementary file 2*). Furthermore, the selective enhancement of feedback inhibitory pattern separation of highly similar inputs was robust over the entire gamma range (up to 100 Hz, *Figure 6—figure supplement 5D,E*).

Next, we tested if our main results were robust to alterations in the relative strengths of feedforward vs. feedback inhibition. Since, our model is closely constrained with respect to the recruitment and functional properties of the feedback circuit, we are confident about the resulting computational inferences concerning this circuit. However, the model does not allow strong inferences about the relative roles of feedback and feedforward inhibition, and it is thus necessary to probe if extremely powerful feedforward inhibition might occlude the effects described here. We therefore selectively enhanced the feedforward inhibitory circuit in our model by increasing the PP to BC circuit 2x (*Figure 6—figure supplement 6*). This robustly increased the feedforward inhibitory contribution to pattern separation above that of feedback inhibition (*Figure 6—figure supplement 6B*). However, it

did not affect the frequency dependence of the feedback inhibitory effect. Indeed, for highly similar input patterns, the feedback inhibitory effect was so prominently enhanced during gamma input, as to again dominate the feedforward inhibitory effect (*Figure 6—figure supplement 6C*).

Finally, we probed the robustness of our findings for various perforant path input strengths (*Figure 6—figure supplement 7*). We found that frequency-dependent pattern separation by the feedback circuit occurred over a large range of PP-input strengths and resulting mean sparsities of the GC population (*Figure 6—figure supplement 7B–D*). These data additionally suggest that for highly similar input patterns, the more efficient sparsification of the GC population at 30 Hz did not fully account for the gains in pattern separation (*Figure 6—figure supplement 7F*). Specifically, selecting a PP-input strength at 10 Hz that produced the same sparsity as during 30 Hz did not allow to reach similar pattern separation (*Figure 6—figure supplement 7F*). This result suggests that the feedback circuit mediates direct assembly competition, allowing pattern separation beyond a pure sparsification effect.

Together these results suggest that frequency dependence is a key feature of the feedback inhibitory microcircuits and predict that feedback inhibition selectively boosts the separation of highly similar input patterns during gamma oscillations.

## Discussion

Across brain regions and species, inhibitory circuits contribute critically to regulating the sparsity and overlap of neural representations (*Cayco-Gajic and Silver, 2019*; *Lin et al., 2014*; *Papadopoulou et al., 2011*; *Stefanelli et al., 2016*). In most, if not all brain regions, feedback inhibition is viewed as important in these capabilities, by directly mediating competition between active cell ensembles (*de Almeida et al., 2009*; *Myers and Scharfman, 2009*; *Rolls, 2010*). In the mammalian DG, feedback inhibition is implemented by an intricate network of interneurons that is capable of delivering spatiotemporally defined inhibition to the principal cell population. How *net* feedback inhibition is functionally organized in mammals, and how it may contribute to pattern separation of biologically relevant, temporally structured input patterns is, however, incompletely understood.

### Quantitative physiological properties of DG feedback inhibition

We have therefore quantitatively described the recruitment of *net* feedback inhibition by defined GC population sizes in space and time in the hippocampal DG, a structure in which sparse activity and inhibition are thought to critically contribute to the function of pattern separation (*Gilbert et al., 2001*; *Hunsaker et al., 2008*; *Leal and Yassa, 2018*; *McHugh et al., 2007*; *Stefanelli et al., 2016*). The proposed role of the feedback inhibitory circuit depends critically on its dynamic range, that is the relation between the number of active principal cells and the resulting feedback inhibition. This property of the feedback circuit is determined by complex, mainly hilar cellular connectivity patterns including the synaptic and intrinsic properties of all participating cells (see e.g. *Espinoza et al., 2018*; *Savanthrapadian et al., 2014*), tabular overview in *Supplementary file 1*). While delving into detailed cell-cell connectivities is clearly important, such studies do not allow the quantitative determination of the gain and dynamic range of *net* feedback inhibition (*Kapfer et al., 2007*; *Silberberg and Markram, 2007*). Using two complementary experimental approaches, we found that *net* feedback inhibition is steeply recruited by sparse populations of GCs (<4%). This is in good agreement with the sparse range of GC activity reported in vivo (*Diamantaki et al., 2016*; *Hainmueller and Bartos, 2018*; *Pernía-Andrade and Jonas, 2014*; *Pilz et al., 2016*; *Schmidt et al., 2012*). In these studies, different time windows were used to define active vs. non-active granule cell populations (one to tens of minutes for electrophysiological, imaging or immediate-early gene studies). The relevant window for assembly competition, however, is much shorter. If we assume random Poisson firing, the electrophysiologically determined rates by *Pernía-Andrade and Jonas (2014)* and *Diamantaki et al. (2016)*, suggest active GCs fractions of <2% for realistic assembly competition time windows of <100 ms. Accordingly, the gain and sensitivity of the circuit are well suited to strongly modulate feedback inhibition within the range of GC activity reported in vivo.

## Frequency-dependent effects of feedback inhibition on pattern separation

In addition to steep recruitment, we have described the temporal and spatial distribution of net inhibition delivered by feedback circuits in the DG. How do these properties influence the pattern separation capability of the dentate gyrus? To address this question, we adapted an established biophysically realistic computational model of the DG circuitry (*Santhakumar et al., 2005*; *Yim et al., 2015*). We first carefully constrained the model to match the spatial and temporal properties of *net* feedback inhibition as assessed in our physiological data. We then fixed all model parameters, and proceeded to probe the ability of this circuit to perform pattern separation on temporally complex oscillatory inputs. The major, highly robust, result of this computational study was that the impact of feedback inhibition on pattern separation is frequency-dependent. Specifically, we find that the separation of input patterns during gamma oscillations > 30 Hz is powerfully and selectively enhanced by the feedback circuit. Remarkably, this mechanism involved decreased interneuron activity and was particularly efficient for very similar input patterns. Such an effect has not been discovered in earlier modeling studies, because most models have discretized time, calculating the pre-inhibition population activity, the resulting inhibition, and the inhibition-corrected population activity in a single time step, sometimes assuming an average corrected population rate within this time step (*Myers and Scharfman, 2009*; *Rolls and Treves, 1998*; *Trappenberg, 2010*). Thus, they do not capture temporal features of feedback circuits. On the other hand, a number of spike based, temporally resolved models have considered only temporally unstructured (Poisson) inputs (*Chavlis et al., 2017*; *Hendrickson et al., 2015*; *Hummos et al., 2014*; *Yim et al., 2015*). We suggest that the precise spatiotemporal organization of the feedback circuit, together with the temporal structure of DG inputs is a crucial determinant of pattern separation. Indeed, the DG and its inputs have a strong, behaviorally relevant, temporal structure (*Lasztóczi and Klausberger, 2017*; *Mizuseki et al., 2009*; *Pernía-Andrade and Jonas, 2014*; *Skaggs et al., 1996*). Novelty experience can induce increased gamma and beta range activity (*Berke et al., 2008*; *Rangel et al., 2015*; *Trimper et al., 2017*), and explorative activity with rearing is also associated with increased gamma oscillations (*Barth et al., 2018*). A previous model has addressed how fast, rhythmic gamma-frequency feedback inhibition may implement a type of 'k-winners-take-all' operation, a basic computational component of pattern separation models (*de Almeida et al., 2009*), although this model relies on faster synaptic timescales than we observed in our compound IPSCs. Perhaps most interestingly, the occurrence of oscillations in the slow-gamma range has recently been reported to be causally related to associative memory formation (*Sasaki et al., 2018*; *Trimper et al., 2017*), a process thought to require pattern separation. Consistent with this finding, *Hsiao et al. (2016)* report DG driven gamma entrainment of CA3, the presumed primary storage location of associative memories. Together, this suggests that the dentate pattern separator may be optimized to rapidly detect subtle degrees of difference within the environment in gamma-dominated exploratory brain states, a capability likely to support successful memory encoding of novel environmental features, and potentially aiding in rapid discrimination during recall.

Importantly, the frequency-dependency of pattern separation was driven by the feedback circuit. This effect was highly robust when varying the decay time constants of the inhibitory synaptic conductances, the time windows of analysis, the similarity measure, or the PP input strength. By contrast, feedforward inhibition and anatomical pattern separation was robustly independent of frequency modulation. Together this suggests that frequency-dependent pattern separation is a key property of the local inhibitory feedback circuit. Importantly, this does not preclude that additional, long range projections may add further complexity (*Szabo et al., 2017*). Also note that in addition to the instantaneous pattern separation mechanisms investigated here, potentially complementary mechanisms at much longer time scales have been proposed involving ongoing neurogenesis (*Aimone et al., 2011*; *Clelland et al., 2009*; *Li et al., 2017*; *Sahay et al., 2011*; *Severa et al., 2017*; *Temprana et al., 2015*).

## Spatiotemporal organization of inhibition and pattern separation

The model also allowed us to examine the impact of the spatiotemporal organization of inhibition on pattern separation. Facilitation of feedback circuits produced a small but robust enhancement of pattern separation, while spatial tuning of feedback inhibition did not. The facilitation of feedback

inhibition is a remarkable feature of the DG, which we to our knowledge have described for the first time. It is in marked contrast to area CA1, where somatically measured feedback inhibition shows strong depression (*Pothmann et al., 2014*; *Pouille and Scanziani, 2004*) and is particularly surprising given the prevalence of depression in the literature on pairwise connections (*Supplementary file 1*). Our experimental and modeling data suggest that the strong facilitation of the mossy fiber input to the feedback circuit is the principal mediator of this net facilitation. Physiologically, facilitation may aid sparse GC spiking to efficiently recruit inhibition, particularly during burst-like activity (*Pernía-Andrade and Jonas, 2014*).

In our model, spatial tuning of feedback inhibition had no effects on pattern separation. This may derive from the fact that PP inputs were spatially broad and random, as suggested by anatomical studies (*Tamamaki, 1997*; *Tamamaki and Nojyo, 1993*). In general, the effect of localized inhibition could be more relevant if synchronously activated populations of GCs are locally clustered (*Feldt Muldoon et al., 2013*). For instance, GCs in the inferior and superior blades of the DG are known to be differentially active (*Alme et al., 2010*; *Chawla et al., 2005*). Accordingly, localized inhibition might be important for independent processing between the two blades. An alternative function of spatially graded inhibition has been proposed by *Strüber et al. (2015)*, who suggest that it is more effective in promoting synchronous gamma oscillations. Accordingly, spatial tuning may play a role in creating the oscillatory dynamics, found here to critically impact the feedback inhibitory pattern separation performance.

In conclusion, this study provides the first comprehensive, quantitative description of the spatio-temporal properties of the DG feedback inhibitory microcircuit, and predicts that these properties will selectively enhance the separation of highly similar input patterns during learning-related gamma oscillations. This mechanism may be relevant for understanding disease states in which there is a coincidence of dentate gyrus-centered pathology with abnormal oscillatory activity, and memory and pattern separation deficits such as temporal lobe epilepsy, Alzheimer's disease or schizophrenia (*Andrews-Zwilling et al., 2012*; *Gillespie et al., 2016*; *Leal and Yassa, 2018*; *Verret et al., 2012*).

# Materials and methods

### Key resources table

| Reagent type (species) or resource | Designation | Source or reference | Identifiers | Additional information |
|---|---|---|---|---|
| Strain, strain background (*Mus musculus*) | C57BL/6N | Charles River | Strain Code 027 | |
| Strain, strain background (*Mus musculus*) | Prox1-Cre | MMRRC-UCD | RRID: MMRRC_036632-UCD | obtained as cryopreserved sperm and rederived in the local facility |
| Strain, strain background (*Mus musculus*) | Ai32-ChR-eYFP | Jackson Laboratory | RRID: IMSR_JAX:012569 | |
| Other | UGA-40 | RAPP Optoelectronics | | Galvanometric, focal laser stimulation device |
| Software, algorithm | Igor Pro 6.3 | Wavemetrics | | |
| Software, algorithm | Python 3.5 scikit learn | *Pedregosa et al. (2011)* | | https://scikit-learn.org/stable/ |
| Software, algorithm | ouropy | Custom Python code. This Paper | | https://github.com/danielmk/ouropy |
| Software, algorithm | pyDentate | Custom Python code, This Paper | | https://github.com/danielmk/pyDentateeLife2020 |
| Software, algorithm | Neuron 7.4 | *Carnevale and Hines, 2006* | | |
| Software, algorithm | Prism 6 | Graphpad | | |

### Animals and slice preparation

All experimental procedures were conducted in accordance to federal law of the state of North Rhine-Westphalia (Aktenzeichen 84–02.04.2014.A254), minimizing unnecessary pain and discomfort. Experiments were performed on horizontal hippocampal slices of 21- to 97-day-old mice. $Ca^{2+}$ imaging and a subset of dual recording experiments were performed in C57/Bl6 mice obtained from

Charles River Laboratories (Wilmington, MA). Optogenetic experiments and the remaining dual recording experiments were performed on double transgenic offspring of Tg(Prox1-cre)SJ39Gsat/ Mmucd, MMRRC Cat# 036632-UCD, RRID: MMRRC_036632-UCD) obtained as cryopreserved sperm and rederived in the local facility (*Gong et al., 2007*; *Gong et al., 2003*) and Ai32-mice (B6;129S-Gt (ROSA)26Sor$^{tm32(CAG-COP4*H134R/EYFP)Hze}$/J, IMSR Cat# JAX:012569, RRID: IMSR_JAX:012569). For preparation the animals were deeply anesthetized with Isoflurane (Abbott Laboratories, Abbot Park, USA) and decapitated. The head was instantaneously submerged in ice-cold carbogen saturated artificial cerebrospinal fluid (containing in mM: NaCl, 60; sucrose, 100; KCl, 2.5; $NaH_2PO_4$, 1.25; $NaHCO_3$, 26; $CaCl_2$, 1; $MgCl_2$, 5; glucose, 20) and the brain removed.

Horizontal 350 μm thick sections were cut with a vibratome (VT1200 S, Leica, Wetzlar, Germany, 300 μm sections for hilar recordings). To obtain maximum-connectivity-plane slices the brain was glued to its dorsal surface (compare *Bischofberger et al., 2006*). The slicing depth at which the temporal pole of the hippocampus first became visible was noted (depth = 0 μm). From here the first four sections were discarded (up to a depth of 1400 μm). The following two to three sections were secured such that one further section before the beginning of the dorsal hippocampus (approximately 2400 μm) could be discarded. Slices were incubated at 35°C for 20 to 40 min and then stored in normal ACSF (containing in mM: NaCl, 125; KCl, 3.5; $NaH_2PO_4$, 1.25; $NaHCO_3$, 26; $CaCl_2$, 2.0; $MgCl_2$, 2.0; glucose, 15) at room temperature. Recordings were performed in a submerged recording chamber at 33–35°C under constant superfusion with carbogen saturated ACSF (3 ml/min). Experiments were performed in the superior blade unless otherwise indicated.

## Electrophysiological recordings

Hippocampal dentate GCs were visually identified using infrared oblique illumination contrast microscopy in a 20x or 60x water immersion objective (Olympus, XLumPlanFl, NA0.95W or Nikon, N60X-NIR Apo, NA1.0W) on an upright microscope (TriMScope, LaVision Biotech, Bielefeld, Germany or Nikon Eclipse FN1, Tokyo, Japan). For IPSC measurements the whole-cell patch-clamp configuration was established with a low chloride cesium-methane-sulfonate based intracellular solution (intracellular solution containing in mM: $CH_3O_3SCs$, 140; 4-(2-hydroxyethyl)−1-piperazineethanesulfonic acid (HEPES-acid), 5; ethylene glycol tetraacetic acid (EGTA), 0.16; $MgCl_2$, 0.5; sodium phosphocreatine, 5; glucose, 10). For GC current clamp experiments a low-chloride solution (CC-intracellular solution containing in mM: K-gluconate, 140; 4-(2-hydroxyethyl)−1-piperazineethanesulfonic acid (HEPES-acid), 5; ethylene glycol tetraacetic acid (EGTA), 0.16; $MgCl_2$, 0.5; sodium phosphocreatine, 5) was used. GCs with input resistances greater than 300 MΩ were discarded in order to exclude immature GCs (*Schmidt-Hieber et al., 2004*). Hilar cells were recorded with intracellular solution containing in mM: K-gluconate, 140; KCL, 5; HEPES-acid, 10; EGTA, 0.16; Mg-ATP, 2; $Na_2$-ATP, 2; pH adjusted to 7.25; 277 mmol/kg without biocytin. 0.3% biocytin (Sigma-Aldrich, B4261). In all imaging experiments and a subset of optogenetic experiments, the intracellular solution additionally contained 100 μM Alexa 594 hydrazide sodium salt (Life Technologies, Carlsbad, USA). The identity of visually and electrophysiologically identified mature GC was confirmed by their dendritic morphology after dye filling in every case tested. Pipette resistance of the patch pipettes was 3–7 MΩ. Voltage-clamp recordings were performed with a Multiclamp 700B (Molecular Devices, Sunnyvale) or a BVC-700A amplifier (Dagan Corporation, Minneapolis). Current-clamp recordings were performed with a Multiclamp 700B. Voltage or current signals were digitized with a Digidata 1322A (Molecular Devices) or (Instrutech ITC-16, Heka Electronics, Ludwigshafen, Germany) at 10 or 50 kHz and recorded using Clampex 10.2 (Molecular Devices) or Igor Pro 6 (Wavemetrics, Lake Oswego) on a PC running Windows XP. All electrophysiological recordings were obtained at least in triplicate, then averaged and counted as a single biological replicate. For IPSC measurements, cells were held at 0 mV including liquid-junction potential correction (estimated at 16 mV). To aid the voltage clamp throughout the cell, this depolarized membrane potential was slowly approached during a 15 min pre-equilibration period, during which Cs$^+$ entered the cell. For CC-recordings liquid junction potential was not corrected. IPSCs were normalized to the maximally elicited IPSC over space and power for each respective cell. Importantly, this normalization does not require prespecification of the location or power at which a respective cell's maximum occurs. Note, that due to this procedure all normalized IPSC values are by definition below 100%. Chemicals for electrophysiological experiments were obtained from Sigma-Aldrich (St. Louis). All drugs were purchased from Tocris Bioscience (Bristol, UK).

## Dual patch experiments

Two GCs within 100 μm of each other were recorded. To test for single GC-induced feedback inhibition 10 to 15 trains of 10 APs at 100 Hz were elicited by brief (3 ms) current injections in one cell. Inhibition was monitored either in VC, while holding the cell at 0 mV to allow the detection of small IPSCs (*Figure 2—figure supplement 3*, n = 7 cell pairs, seven directions) or current clamp while holding the cell at −60 mV, allowing to probe for inhibition in both directions (not shown, n = 4 cell pairs, eight directions).

## Ca$^{2+}$ imaging

Dye loading was modified from *Garaschuk et al. (2006)* and performed in the submerged chamber at 35°C under constant superfusion. Briefly, a dye solution containing: 1 mM Oregon Green 488 BAPTA-1 acetoxy-methyl ester (OGB-1 AM); 2% pluronic F-127; 150 mM; 2.5 mM KCl; 10 mM HEPES). The dye was injected into the slice along the superior blade of the GC layer using standard patch pipettes (4–5 locations, 100 μm intervals, 30 μm depth, 3 min at 500 mbar per location). Recordings were started at least 45 min after the staining procedure. Population Ca$^{2+}$ Imaging was performed using a multibeam two-photon fluorescence microscope (TriMScope, LaVision Biotech, Bielefeld, Germany) with excitation light at 810 nm. Images were acquired with a digital CMOS camera (ORCA-Flash, Hamamatsu) through a high numerical aperture 20x water immersion Objective (XLumPlanFl, NA-0.95, Olympus). This allowed imaging of a large field of view (320 × 240 μm) with high spatial and temporal resolution (1920 × 1440 pixels, 20 Hz) at acceptable signal to noise ratios. Time series were processed with ImageJ 1.48o and IGOR Pro 6.3 in a semiautomatic manner. Regions of interest were manually placed onto all well loaded cells which remained visible throughout the experiment. Ca$^{2+}$ fluorescence increase normalized to baseline (ΔF/F) traces of individual cells were calculated without background subtraction. The fraction of responders for each time series was extracted by automatic thresholding at ΔF/F = 0.94%. The threshold was determined by combined cell-attached and Ca$^{2+}$ imaging experiments. Note, that for these experiments the stimulation electrode was placed into the hilus in order to obtain a sufficient number of true positive responders. The imaged cell population comprised on average 46 ± 18 (standard deviation) cells (n = 23 slices). The active cell fraction corresponds to the fraction of responders normalized to the dye-loaded population within each section. To assess the spatial distribution of cell activation in imaging experiments, ΔF/F projections were created by averaging and smoothing four frames during the transient and four frames at baseline fluorescence and then calculating the pixel wise ΔF/F.

Antidromic electrical stimulation was achieved using a bipolar cluster microelectrode (FHC, Bowdoin) connected to a digital stimulus isolator (AM-systems, Sequim), placed into stratum lucidum in the CA3 region. IPSCs at individual powers were elicited 5 to 13 times at 0.1 Hz and averaged (0.1 ms pulse time). The amplitude beyond which the stimulus isolator could not pass the full current, determined the maximal stimulation amplitude for each experiment.

In order to obtain the input-output relationships of the feedback inhibitory circuit data, each variable was averaged over slices by power. This was necessary since only a small subset of experiments in which inhibition was completely blocked could also be successfully imaged (6 of 8 sections). Due to the small numbers of active cells within individual slices with sufficient dye loading (n = 23 slices) analysis of only these six slices leads to a very piecemeal recruitment curve. A more accurate estimation of the recruitment of feedback inhibition was obtained by averaging the cell activation and inhibition over all appropriate slices and relating them by power, respectively. Note that while the fraction of activated cells in non-MCP sections (not included in the quantitative analysis) was mostly zero, IPSCs were almost always present (in 28 of 29 cells in non-MCP sections).

## Optogenetic stimulation

Focal optogenetic stimulation was achieved through a galvanometer driven spot illumination device coupled to a 473 nm DPSS Laser (UGA-40, DL-473, Rapp Optoelectronics, Hamburg, Germany) on an upright microscope (Nikon Eclipse FN1, Tokyo, Japan). The width of the resulting stimulation spot at the focal plane was 8.36 ± 0.04 μm (full width at half max; Nikon 10X Plan Fluor, NA 0.3 Laser powers are given in arbitrary units from 1 to 7 corresponding to 15 ± 1 μW, 107 ± 14 μW, 292 ± 42 μW, 762 ± 105 μW, 1433 ± 49 μW, 1729 ± 165 μW and 1660 ± 163 μW at the objective (n = 5

measurements). All illumination spots were placed at approximately 40 µm into the ML at the slice surface. Stimulation pulses were of 20 ms duration.

## Light intensity distribution

To measure the light intensity distribution throughout a slice the setup was modified to image the slice from below while the laser beam was focused to its surface (*Figure 2—figure supplement 1C–F*). This was achieved by focusing a surgical Microscope with 36x magnification (M695, Leica Microsystems, Wetzlar, Germany) to the lower slice surface. Images were taken with a CCD camera (Nikon D60). Acute sections of 100, 150, 200, 250, 300 and 350 µm thickness were cut from Prox1-ChR-eYPF mice as described above. The laser was focused to the surface of the slice in the molecular layer and an image was taken at every laser power (p=1 to 7 AU). The stage was moved for every image to avoid bleaching or phototoxicity. Linear profiles of the resulting isometric light distribution were measured in several directions and averaged to obtain an x profile per section. The x-profiles of slices of different thickness were then stacked to obtain the xz-profile. Values below 100 µm depth were obtained through fitting a Gaussian function in x-direction at 100 µm depth and an exponential function in z-direction. Complete three-dimensional intensity profiles of three different locations of two slices within the dentate molecular layer were averaged.

## Calculation of the optogenetically activated cell fraction

To assess the active fraction of GCs, approximately two GCs were recorded in cell-attached mode in each slice in which an IPSC was recorded. Illumination spots were placed along the GC layer at 100 µm intervals (*Figure 2—figure supplement 1*). The entire profile was probed in triplicate with 1 s intervals between individual locations. When the stimulation spot was in sufficient proximity to the recorded cell clear APs were generally visible (in 25 of 26 cells), and otherwise could be induced through simultaneous cell attached depolarization. Cell-attached spikes were detected by automatic thresholding at 6x standard deviation of the baseline. The spatial profile of firing probabilities, centered on the recorded cells, was averaged within each section. To test if cell activation properties differed between blades the maximum firing probabilities (at p=7) as well as the slopes (increase in firing probability from p=1 to 7) when simply averaging over all location of a given cell were compared by t-test (n = 7 sections per blade, p=0.490 and 0.684 for max. AP probability and slope, respectively). Since no difference was observed a single firing probability distribution as a function of the distance along the GC layer (x – distance) was calculated for each power (*Figure 2—figure supplement 1B*, n = 14 sections, seven per blade). However, the firing probability of cells in the vicinity of the illumination spot is likely to increase not only as a function of the laser power and spread at the surface, but also of the penetration depth of the light cone. In order to calculate the firing probabilities throughout the slice, the firing probability distribution at the surface was related to the measured light intensity distribution throughout the slice (*Figure 2—figure supplement 1C–F*; see above) utilizing a 'virtual distance' measure. Since cells were measured at random distances from the molecular layer border, the light intensity distribution, like the firing probabilities were collapsed to two dimensions, x-distance along the GC layer and z-distance with increasing slice depth. The 'virtual distance' was calculated as the mean distance from a given slice-surface pixel to all other pixels of the light intensity distribution weighted by the intensity within those pixels (*Figure 2—figure supplement 1G*). Assigning the firing probabilities of pixels at the slice surface to their respective virtual distance yields the firing probability distribution as a function of virtual distance, which was well approximated by a gaussian fit (*Figure 2—figure supplement 1H*). This fit was used to also calculate the firing probabilities of pixels/cells deeper in the slice using the measured light intensity distribution as input. The active cell fraction then corresponds simply to the mean firing probability throughout the slice. This calculation is independent of the size and number of GC and was performed for every power individually. We noted that a large fraction of the recorded spikes occurred with larger latency than the typical IPSC following the beginning of the 20 ms stimulation pulse (*Figure 2—figure supplement 1I*, example from a single slice). Since only APs preceding the IPSC can participate in its recruitment, we calculated the fraction of total spikes which preceded mean IPSC latency for every power, and fitted the resulting relation with an exponential function (*Figure 2—figure supplement 1J*). All active cell fractions were corrected by this factor (*Figure 2—figure supplement 1J*, bottom). Note that this does not take account of the disynaptic delay between mossy fiber output

and GC input, thereby potentially slightly overestimating the true recruiting population. For comparison, the active cell fraction was also computed with alternative assumptions about the decay of the firing probability with increasing slice depth. If no firing probability decay with increasing depth is assumed, the active cell fraction throughout the slice is given simply by the average of the measured firing probabilities at the slice surface (*Figure 2—figure supplement 1K*, upper grey dashed line). Alternatively, the firing probability decay with depth was assumed to be identical to the measured decay along the slice surface (isometric firing probability distribution; *Figure 2—figure supplement 1K*, lower grey dashed line). In this case, Gaussian functions were fit to the probability distributions at the surface and these Gaussian functions were then assumed to extend also in the z-dimension. The active GC fraction was then calculated by numerical integration under the two dimensional Gaussian (with the bounds from 0 to 350 µm in z and −888 to 888 µm in x, which corresponds to the mean GC layer length) normalized to the same area with a uniform firing probability of one. The best estimate of the active GC fraction, incorporating light intensity measurements (*Figure 2—figure supplement 1K*, black line), was within these upper and lower bound estimates.

## Comparison of focal and global activation

To globally activate the GC population a multimode light fiber (BF-22, Thorlabs, New Jersey) coupled to a 473 nm laser (Omicron Phoxx, Rodgau-Dudenhofen, Germany) was placed above the slice surface, non-specifically illuminating the entire hippocampus. Analogous to focal stimulations, the activated cell fraction was calculated as the firing probability of individual cells following 20 ms pulses. Here, no spatial normalization is necessary since cells were sampled from random locations with respect to the light fiber. Firing probabilities for the focal stimulation in these sections was calculated as the simple average of all stimulation locations.

## Spatial distribution of feedback inhibition

The same stimulation paradigm which was used to assess cell activation was used to assess the spatial distribution of feedback inhibition. For individual cells, IPSCs at each location and power were averaged. The entire profile was normalized to the largest measured IPSC of that cell, independent of the power and stimulation location at which it occurred. For analysis, all IPSC profiles were spatially aligned to the recorded cells. The mean distance to apex ± one standard deviation was 356 ± 163 µm and 322 ± 97 µm for cells from the superior and inferior blade, respectively (n = 8 cells in each blade). In order to test whether there were any distinct effects of the apex, such as a steep decay of inhibition, which would be masked by alignment to the recorded cells, we also aligned the profiles to the apex (not shown). However, no such effects were visible. To analyze the saturated IPSC profiles, normalized IPSC amplitudes from p=5 to 7 were averaged for each cell. In order to analyze the effects of local versus remote stimulation for each blade a distance was chosen such that each remote location was still within the DG but in the other blade (800 µm from the recorded cell). Normalized IPSCs of the three locations surrounding the recorded cell or this remote location were averaged within each power to obtain the IPSC amplitudes for further analysis. The cell fraction required for the activation of a half-maximal IPSC in each section was assessed for each cell by linear interpolation between the measured values. Since no differences were found between superior and inferior inhibition, recordings of both blades were pooled to analyze the kinetic properties of IPSCs. All parameters were calculated on the multiple trials of individual cells. The latency was measured as the time from the beginning of the pulse to when the IPSC superseded six fold standard deviation of the baseline. The jitter was calculated as the standard deviation of these latencies for individual cells. The rise time was calculated as the mean 20 to 80 rise time of each cell and the decay time constant was obtained from an exponential fit to the decaying phase of the compound IPSC.

### Hilar recordings

Intrinsic properties of hilar cells were quantified based on 4.6 s long depolarizing current steps or 500 ms hyperpolarizing current steps. AP threshold and fast AHP amplitude were measured from the first AP in the first current step in which an AP occurred within the first 10 ms. Clustering fraction and mean AP time were calculated from the current injection that elicited the maximum average AP frequency. The Clustering fraction represents the fraction of APs that occur within 60 ms before or after another AP (*Larimer and Strowbridge, 2008*). Mean AP time was calculated as the mean AP

time point normalized to the duration of the current injection (4.6 s). Input resistance was calculated as the slope of the IO curve from the hyperpolarizing current ladder. Cells were manually classified as mossy cells or interneurons based on these intrinsic properties. To objectively confirm classification, we performed unsupervised k-means clustering using scikit-learn (*Pedregosa et al., 2011*). For clustering all six measures were normalized by mean and variance. Two cells with conflicting classification were not included in further analysis.

After recording, slices were fixed for 1 hr in 4% PFA and stored overnight in 0.25% PBS-T at room temperature. The following day they were transferred to PBS for short term storage or immediately stained. For biocytin staining, sections were washed with PBS and incubated with Streptavidin-Alexa-Fluor-555 Conjugate (Invitrogen, S32355), 1:1000 in 0.25% PBS-T overnight at 4°C. The following day they were co-stained with DAPI 1:1000 in PBS for 30 min and mounted with Aqua-Poly/Mount. Cells were imaged with the Leica SP8 Confocal Microscope of the Microscopy Core Facility at the University Clinic Bonn using a 40x water immersion objective.

## Short-term dynamics

Short-term dynamics of GCs and hilar cells were assessed using antidromic electrical or optogenetic stimulation at minimal power (the smallest stimulation power that yielded reliable responses). Trains of 10 pulses at 1, 10, 30, 50 Hz were delivered in triplicate and averaged (excluding sweeps with action currents for hilar cells). In all GCs and a subset of hilar cells we confirmed that PSCs could be blocked by at least 90% with 40 μM CNQX + 50 μM D-APV (n = 12, 23 for GCs and hilar cells respectively). Facilitation indices were obtained by normalizing the average of the last three PSC peaks to the first.

To test for differential dynamics between local and remote inhibition analogous trains of optogenetic 20 ms pulses at powers below saturation (usually p=2 for local inhibition and p=3 for remote inhibition) were delivered. For each power and frequency, five repeats were recorded and averaged. AP probabilities were assessed by cell-attached recordings with the stimulation site close to the recorded cell. Cell-attached spikes were detected by automatic thresholding as above.

## Voltage escape estimation model

A simple multicompartmental passive 'ball and stick' model with number of segments following the d_lambda rule (*Carnevale and Hines, 2006*) and passive properties Ra = 181 Ωcm, Cm = 1 uFcm$^{-2}$ and a leak conductance = 0.0002 Scm$^{-2}$, which gave an Rin of 165 MΩ, were adopted from *Carnevale and Hines (2006)* and *Krueppel et al. (2011)*. A soma (20 μm diameter) contained one dendrite (3 μm diameter, 200 μm length) with an alpha synapse point mechanism (Erev −90 mV) placed at 180 μm from the soma. The range of synaptic conductances (0.1–50 nS; adopted from *Williams and Mitchell, 2008*) elicited IPSC amplitudes in the model, which covered the range of somatic IPSC amplitudes that were experimentally measured (3 pA – 1nA). Voltage clamp experiments were simulated using a single electrode point mechanism at the soma (Rs 5 MOhms, to model a Rs of 15 MΩ compensated 70%) with a holding potential of 0 mV. The transfer (Zc) and input impedance (Zn) were determined from the model and used to calculate the actual peak IPSC amplitude at the soma for a given synaptic conductance. Simulations were run in the Neuron 7.5 simulation environment.

## Biophysically realistic dentate gyrus lamella model

Simulations were run in python 2.7 with NEURON 7.4 (*Carnevale and Hines, 2006*) on Windows 7/10. We created a generic python-NEURON interface (https://github.com/danielmk/ouropy; copy archived at https://github.com/elifesciences-publications/ouropy) which wraps NEURON's python module, into which we ported the conductance based DG model by *Santhakumar et al. (2005)*. Model code is available at https://github.com/danielmk/pyDentateeLife2020 (copy archived at https://github.com/elifesciences-publications/pyDentateeLife2020).

We first tuned the original model to capture our experimentally determined properties in the most parsimonious way. During tuning we also updated some model properties to better reflect current data and our experimental paradigm in an individual DG lamella:

We introduced a T-type Ca$^{2+}$ channel mechanism into MCs to more realistically reflect the depolarizing envelope at the onset of a positive current step observed in real MCs. Furthermore, while

the original model placed the perforant path input at the distal dendrite of GCs, we moved all perforant path synapses to the middle compartment of the dendrite. In order to be able to capture the results of convergent and divergent synaptic inputs in sufficient resolution to produce the empirically observed activity gradations, we up-scaled cell numbers by a factor of four. To model space, we assumed all cell types to be spread out on a 2 mm DG lamella. Since MCs project to GCs primarily outside the lamellar plane, we removed the MC to GC connection. To allow patterned PP input we adapted PP input specifications from *Yim et al. (2015)*.

We then proceeded in a first phase of model adjustment, and adapted several parameters to reproduce our in vitro findings regarding spatial and temporal feedback inhibition (*Supplementary file 2*). To model frequency-dependent facilitation on mossy fiber outputs, we implemented a simple frequency-dependent synapse model (tmgsyn) (*Tsodyks et al., 1998*), and matched the facilitation time constant as well as the decay time constants of individual PSCs to our experimental observations. As in the original model, each cell gives rise to a fixed number of synaptic connections which are spatially restricted to a target pool of adjacent cells. We tuned the size and spatial extent of this target pool to reproduce our spatial data. To provide local inhibition we implemented a 'local' interneuron type (BC), whose inputs and outputs were spatially restricted to an ~600 µm area (as described by *Strüber et al., 2015*). To provide global inhibition we implemented a second class of inhibitory interneurons (HC) whose inputs and outputs connect to GCs independent of space. This simple formulation allowed us to reproduce the recruitment curves seen for local, remote and global GC activation paradigms. To achieve plausible activity levels, we further adapted synaptic weights similar to *Yim et al. (2015)*. We call the network incorporating both spatially restricted BC synapses and mossy fiber facilitation the full tuned network. To isolate the contribution of intrinsic GC properties to pattern separation, we created a disinhibited network by setting the synaptic weight from all interneurons to zero. We also isolated feedforward inhibition by decreasing the mossy fiber to interneuron synaptic weight to zero. To evaluate the effect of spatially constrained inhibition, we created a global network, where the target pool of all interneuron was the entire GC population. To evaluate the effect of mossy fiber facilitation, we set the facilitation time constant to zero, effectively eliminating facilitation. Details on the model parameters are summarized in *Supplementary file 2*).

To study pattern separation, we generated 400 PP inputs. Each PP synapsed onto 100 randomly chosen GCs with the spatial connection probability being governed by a gaussian probability distribution with standard deviation 1 mm and random peak position, modeling a full, nearly uniform input connectivity of individual afferents (*Tamamaki and Nojyo, 1993*). To generate theta modulated spike patterns, we used the inhomogeneous Poisson generator from Elephant 0.5.0-Electrophysiology-Analysis-Toolkit with a 10 Hz (theta) sinusoidal rate profile with a peak of 100 Hz, a minimum of 0 Hz and a duration of 600 ms. To generate input patterns with varying overlap from PP afferents i = 1 to 400, we activated afferents i to i+23 in increments of i = 1 per run. We performed 25 runs for each condition resulting in 300 unique comparisons, excluding self-comparisons. The random seed was held constant between different runs of the same condition, resulting in differing input patterns being fed into the same network. All randomness was generated with the python module numpy.random.

To quantify pattern similarity, we used Pearson's product moment correlation coefficient R of the population rate vectors for input and output patterns. The population rate vector refers to the vector of the mean firing rates of all cells in the population within the entire 600 ms simulation, or 100 or 33 ms time windows for the time resolved analyses. All statistical analyses of the model were performed with n = 7 different random network seeds. During Model development (tuning phase), we first ported the model by *Santhakumar et al. (2005)* with closely constrained DG cell-types, and further constrained it to reproduce our physiological data. We then locked the model and proceeded to an (in silico) experimental phase, in which pattern separation was investigated.

To compute full pattern separation effects (*Figure 6D*), we calculated the mean $R_{out}$ within $R_{in}$ bins of 0.1 and measured the area to the unity line (computed as the mean of the binwise $R_{in} - R_{out}$ differences). To compute isolated pattern separation effects of specific manipulations we subtracted the respective $R_{out}$ values with and without the manipulation, thereby obtaining a $\Delta R_{out}$ value for each individual $R_{in}$. We then again computed the bin-wise mean and quantified the area under the curve, yielding the mean $\Delta R_{out}$ analogous to the full effects. Note, that the sequence of averaging and subtracting is irrelevant, and was inverted only to match the figure panels. Data are displayed as

mean ± SEM for each $R_{in}$ bin (*Figure 6E–G*). The coefficient of variance (CoV) was calculated by normalizing the standard deviation of $\Delta R_{out}$ within each bin by the mean of that bin, and then averaging over bins, analogous to the previous analyses. However, only bins within $0.2 < R_{in} < 0.8$ were included, since at the borders very small means led to unreliable results. $\Delta$CoV represents the difference between the mean CoV of the global (or nonfacilitating) and the tuned network models. For the temporally resolved pattern separation analysis, all measures were computed as above, but on population vector correlations within 100 or 33 ms time bins.

## Statistics and Data Analysis

Analyses were performed using ImageJ, Microsoft Excel, Python and Igor Pro. Fits were performed using Igor Pro. Statistical analyses were performed using GraphPad Prism six or Igor Pro. Comparisons were two-tailed whenever applicable. Replicates refer to cells unless otherwise indicated (slices for imaging experiments and network seeds for modeling data). Given the lack of previous information on effect sizes, sample sizes were chosen according to field norms, such that only large effects can be detected (e.g. Cohen's d > 1 for paired tests). A single outlier facilitation index (*Figure 5E*) during model tuning was removed, as it was outside the triple standard deviation (due to a very small initial IPSC). Group allocation was achieved by alternating acquisition between groups. Statistical significance in Analysis of Variance (ANOVA) is indicated by §. F-values and degrees of freedom are given as F(DFn, DFd). When ANOVAs were followed by specific comparisons these are indicated by asterisks, where *p<0.05, **p<0.01 and ***p<0.001. Bargraphs and XY plots show means where error bars indicate standard error of the mean. In boxplots error bars represent the data range and boxes the upper and lower quartiles and the median.

## Acknowledgements

This work was supported by the Deutsche Forschungsgemeinschaft (SFB 1089, ebGluNet, SPP2041), the BONFOR program of the University of Bonn Medical Center, the ERANET Neuron grant 'EpiNet' (to HB). We thank Olivia van Ray and Dominik Holtkamp for excellent technical assistance; Thoralf Opitz, Holger Dannenberg, Laura Ewell and Dirk Dietrich for valuable comments and Stefan Remy for access to multi-beam two-photon microscopy. We further acknowledge support by the Microscopy Core Facility of the Medical Faculty.

## Additional information

### Funding

| Funder | Grant reference number | Author |
| --- | --- | --- |
| Deutsche Forschungsgemeinschaft | SFB1089 | Heinz Beck |
| Deutsche Forschungsgemeinschaft | ebGluNet | Heinz Beck |
| European Commission | EpiNet | Heinz Beck |

The funders had no role in study design, data collection and interpretation, or the decision to submit the work for publication.

### Author contributions

Oliver Braganza, Conceptualization, Resources, Data curation, Software, Formal analysis, Investigation, Visualization, Methodology; Daniel Mueller-Komorowska, Conceptualization, Resources, Software, Validation, Investigation, Methodology; Tony Kelly, Software, Supervision, Investigation, Methodology; Heinz Beck, Conceptualization, Resources, Supervision, Funding acquisition, Project administration

### Author ORCIDs

Oliver Braganza  https://orcid.org/0000-0001-8508-1070
Daniel Mueller-Komorowska  https://orcid.org/0000-0002-2789-6068

### Ethics

Animal experimentation: This study was performed in strict accordance with the recommendations of the Landesamt für Natur, Umwelt und Verbraucherschutz Nordrhein-Westfalen (LANUV, Aktenzeichen 84-02.04.2014.A254). All animals were anesthetized with isoflurane prior to euthanasia and organ extraction.

### Decision letter and Author response

Decision letter https://doi.org/10.7554/eLife.53148.sa1
Author response https://doi.org/10.7554/eLife.53148.sa2

## Additional files

### Supplementary files

• Supplementary file 1. Literature review for DG circuit short-term dynamics. Studies reporting short-term dynamics within the DG circuit were reviewed with a main focus on facilitation or depression of synaptic connections defined by pre and postsynaptic cell types. Note the abundance of depressing synapses (quantitative descriptions of depression blue). Also note the complexity of direct connections between Interneurons (lower third of the table).

• Supplementary file 2. Model parameters. Overview of synaptic and intrinsic parameters between model cell-types. First row includes modeled cell number per type. **PP**: perforant path, **GC**: granule cell, **MC**: mossy cell, **BC**: basket cell, **HC**: Hilar perforant path associated cell; Weight: maximal synaptic conductance, Facilitation Max.: maximal fold increase of synaptic conductance, Decay Tau: synaptic decay time constant, Facilit. Tau: facilitation time constant, Delay: latency to postsynaptic event after presynaptic action potential, Target pool: range of n closest cells potentially receiving an output, Divergence: number of output synapses per cell stochastically picked from target pool, Target segments: cellular compartment receiving the synapse. Values in brackets are values for robustness analyses in Figs. S10, S11.

• Transparent reporting form

### Data availability

The generic python-NEURON interface is available at https://github.com/danielmk/ouropy (copy archived at https://github.com/elifesciences-publications/ouropy). The model code is available at https://github.com/danielmk/pyDentateeLife2020 (copy archived at https://github.com/elifesciences-publications/pyDentateeLife202).

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
