## [Decision Letter]

**Acceptance summary:**

This study utilizes electrophysiological, optogenetic and computational modeling approaches to provide comprehensive and quantitative description of the spatiotemporal properties of the dentate gyrus feedback inhibitory microcircuitry. It predicts that these properties selectively enhance the separation of highly similar input patterns during learning- related gamma oscillations. The data were found to be high quality, and the idea that pattern separation is frequency dependent will likely be of interest to many researchers.

**Decision letter after peer review:**

Thank you for submitting your article "Quantitative properties of a feedback circuit predict frequency-dependent pattern separation" for consideration by *eLife*. Your article has been reviewed by three peer reviewers, and the evaluation has been overseen by a Reviewing Editor and Laura Colgin as the Senior Editor. The following individual involved in review of your submission has agreed to reveal their identity: Mathew V Jones (Reviewer #2).

The reviewers have discussed the reviews with one another and the Reviewing Editor has drafted this decision to help you prepare a revised submission.

Summary:

This study examines the recruitment of feedback inhibitory interneurons by dentate gyrus (DG) granule cells (GCs) using single whole-cell recordings from GCs during extracellular stimulation of mossy fiber axons in CA3 or optogenetic stimulation of GC populations. The authors very elegantly demonstrate that feedback inhibition in the DG is recruited in a low dynamic range of 0-4% of the active GC population. Moreover, the study demonstrates that inhibition is not uniformly distributed but inhibitory signals show a larger amplitude, low jitter of individually evoked IPSCs and faster time course when induced close to the recorded GC than inhibitory signals evoked at more remote sites in the DG. On the basis of the experimental data and a neuronal network model consisting of the various neuronal components of the DG, the authors demonstrate that such a network is suited to perform pattern separation between two overlapping inputs provided by the modelled perforant path particularly at gamma frequencies. The experimental and computational work is very well performed and the data are of high interest to the currently broadly discussed potential role of the DG in pattern separation. However, addressing the following issues would enhance the strength of the conclusions.

Essential revisions:

1) "Feedback inhibition is recruited steeply with low dynamic range (0-4% of activate GCs)". This central conclusion is very interesting but not fully convincing.

In Figure 1, only 4% of GCs are activated by antidromic stimulation (Figure 1H) which subsequently leads to the 4% ceiling shown in Figure 1K. But the critical parameter may not be how many GCs are stimulated but rather how many MFs are stimulated, since MFs that are unconnected to GCs likely contribute to feedback inhibition (i.e. Figure 1I shows that 30% max of feedback IPSC is recruited by 0 active GCs). Whether the low fraction of GCs responding to antidromic stimulation reflects low connectivity in the slice or inability to recruit MFs is unclear, but either way it raises a question about the robustness of this assay because this limitation dictates the upper limit for% active GCs. Reviewers wondered if Ca^2+^ imaging of the MFs near the stimulation site would provide a better dynamic range to make a quantitative estimate of% active MFs.

Figure 2. Reviewers appreciated that the authors use a 2nd method to estimate feedback inhibition wherein the full range of MF/GC activation can be assessed. Reviewers found it very surprising that the max IPSC evoked by focal and global Chr2 stimulation was similar (Figure 2I), and that the range of max IPSCs was so variable (<0.1 to >0.8 nA). What is the fraction of hilar interneurons recruited by focal and global ChR2 activation? This might help explain this surprising result and the variability. The authors state (subsection “Quantitative physiological properties of DG feedback inhibition”) that the dynamic range is determined by the cellular connectivity patterns, but it is not clear how connectivity could generate this narrow dynamic range – please explain. In measuring feedback IPSCs, the authors addressed voltage escape for dendritic IPSCs (Figure 2—figure supplement 2), but they did not address whether the ChR2-mediated conductance evoked by global light activation affects the measurement of the global max IPSC. In addition, why do the normalized curves not reach 100% (Figure 2 D and H)? Same question for all other figures where "IPSC [%]" is reported.

2) Figure 3. The slow rise time of IPSCs seem inconsistent with PV interneurons that are thought to be the major source of feedback inhibition. Presumably the IPSCs are slowed by asynchrony/compound events which might be assessed by quantifying individual events rather than averages to infer PV involvement. Were any PV interneurons recruited in Figure 4? It seems important to address the contribution of PVs since they are critical interneuron subtype in the model. Based on reported uIPSC from PVs, can the authors estimate how many PVs might be contributing to feedback inhibition?

3) The small number of active PP afferents used (24 of 400) in the input patterns meant that only about 10% of GCs were active even in the absence of inhibition (Figure 6—figure supplement 2). Just as the authors tested the robustness of their conclusions in the face of stronger feed-forward inhibition, it also seems important to assess the robustness of the conclusions across a range of excitatory drive that might represent EC activity under various conditions.

4) Subsection “Quantitative physiological properties of DG feedback inhibition”. The authors report that the range of active GCs that saturate feedback inhibition is in the range of active GCs reported in vivo. However, Pernia-Andrade and Jonas, 2014, Pilz et al., 2016, and Diamantaki et al., 2016, report higher fractions of active GCs in vivo, although reports using cFos labeling typically support the 1-4% range. The authors should discuss more explicitly the literature about GC activity in vivo and the interpretations of their small dynamic range of feedback inhibition in light of the possibility that GC activity might not be as sparse as suggested by cFos.

5) The nature of the facilitating inhibition remains unclear. Fast-spiking basket cells in the hippocampus usually show paired pulse or multiple pulse depression and for only few interneuron types paired pulse or multiple pulse facilitation has been observed. Whole-cell recordings from GCs during optogenetic stimulation of the different types of DG-interneurons might help to dissect the nature of the facilitating inhibitory inputs (e.g. Somatostatin-expressing cells).

6) GCs also contact Mossy cells (MCs), which in turn recruit DG interneurons and thereby inhibit GCs. It remained unclear why the MC feedback excitation of hilar interneurons was removed from the network model. It is an important functional element of this circuitry, which provides inhibition to GCs.

7) Subsection “Input-output relation of the feedback inhibitory microcircuit”, second paragraph and Figure 1—figure supplement 1 legend – Is it correct that the detection threshold used (0.94%) leads to a "true positive rate of 3%"? That seems very low, and implies that 97% of true responses were not detected. Unless this is a typo, is this not a serious problem that implies that the estimates of the active fraction of GCs are extreme underestimates?

8) Frequency-dependence of facilitation – Please state explicitly whether there was a "frequency tuning" (e.g., a preferred frequency) or whether all frequencies {greater than or equal to} 10 Hz displayed the same facilitation ratio (greater than 1).

9) Spike rates – Here, pattern separation was computed from Pearson's correlation, dot product and pattern overlap of population spike rate vectors, all of which are in general sensitive to absolute spike rates. Therefore, it is not surprising that some of the manipulations in the model (e.g., reducing various sources of inhibition) would enhance correlations by increasing GC spike rates, thus reducing pattern separation. It would be extremely useful to know if the different outcomes in pattern separation are driven mainly by the impact of the various sources of inhibition on GC spike rate alone, or whether there is indeed something else "special" about the different sources of inhibition, such as their differential IPSC latencies following PP or GC spikes, their IPSC timing jitter or failure rate or their input location onto the GCs. Plots of average GC spike rate versus the area under the ∆R_out_ curve, for the seven model families chosen, would be a first-pass at addressing these questions.

10) "The effect of facilitation on pattern separation is intuitive, since this allows the feedback circuit to integrate GC activity over time, and convert it to inhibition." Reviewers were not sure how intuitive this really is. True, integrating GC activity over time might be useful, but a) depression would allow a mathematical differentiation (or high-pass filtering) of GC activity over time, which arguably could be better than integrating over time for the purpose of separating temporal patterns, and b) the output of the inhibitory circuit is largely depressing anyway, providing the differentiation mentioned above. Thus I think the question remains: what is the purpose of the facilitation of the GC->BC synapse if the ultimate output will be depression at the BC->GC synapse anyway. Indeed, the authors found that the GC->BC facilitation only had a small effect on pattern separation.

---

## [Author Response]

Essential revisions:1) "Feedback inhibition is recruited steeply with low dynamic range (0-4% of activate GCs)". This central conclusion is very interesting but not fully convincing.In Figure 1, only 4% of GCs are activated by antidromic stimulation (Figure 1H) which subsequently leads to the 4% ceiling shown in Figure 1K. But the critical parameter may not be how many GCs are stimulated but rather how many MFs are stimulated, since MFs that are unconnected to GCs likely contribute to feedback inhibition (i.e. Figure 1I shows that 30% max of feedback IPSC is recruited by 0 active GCs). Whether the low fraction of GCs responding to antidromic stimulation reflects low connectivity in the slice or inability to recruit MFs is unclear, but either way it raises a question about the robustness of this assay because this limitation dictates the upper limit for% active GCs. Reviewers wondered if Ca^2+^ imaging of the MFs near the stimulation site would provide a better dynamic range to make a quantitative estimate of% active MFs.

The reviewer is absolutely right that the critical parameter will be the number of (sufficiently conserved) mossy fibers stimulated, and that our imaging approach only captures a sample of the population (that within the imaging plane). Clearly, in the example from an individual slice (Figure 1I), there are activated cells or axons outside the imaging plane. For this reason the average across slices (Figure 1J), where 0.36% of GCs lead to 12% inhibition (50µA stim), is a more appropriate estimate. Particularly the observation, that the maximal population fraction imaged (~4%) precludes inferences about higher cell fractions, is well taken. We therefore only cautiously stated ‘a first quantitative estimate’. Indeed, these caveats were the reason why we proceeded to the optogenetic approach, which was not limited by the electrical stimulation technique in terms of how many GCs could be activated, and is not subject to the potential underestimation error pointed out by the reviewer.

Concerning the reviewer’s suggestion to image MFs near the stimulation site, we are unsure if this would yield much better estimates. Firstly, any imaging plane would inevitably also only detect a sample of the activated mossy fibers. Secondly, the smaller size of axons, compared to somata, would likely imply substantially worse signal to noise. Thirdly, there would be the added complication that multiple imaged axon collaterals within our image plane might be counted as separate cells. Fourthly, counted active axons could be cut within stratum lucidum, i.e. prior to the hilus. Because of these caveats we reasoned that the number of active somata would in fact allow a reasonable estimate of the number of activated, well-conserved, individual mossy-fibers.

Therefore, we believe the reviewers concern, while absolutely valid in the context of the Ca²^+^-imaging experiments, is in fact best addressed by the subsequent optogenetic experiments. These demonstrate clearly, that activating cell fractions >4% does not lead to the recruitment of substantial further inhibition.

Figure 2. Reviewers appreciated that the authors use a 2nd method to estimate feedback inhibition wherein the full range of MF/GC activation can be assessed. Reviewers found it very surprising that the max IPSC evoked by focal and global Chr2 stimulation was similar (Figure 2I), and that the range of max IPSCs was so variable (<0.1 to >0.8 nA). What is the fraction of hilar interneurons recruited by focal and global ChR2 activation? This might help explain this surprising result and the variability.

For global activation, we know from Figure 2I that this stimulation reliably activates 100% of granule cells. Given the recruitment curve of feedback inhibition, it is extremely likely that this form of stimulation recruits 100% of interneurons within the feedback circuit to generate action potentials. Even so, global stimulation recruited IPSCs of variable sizes (Figure 2I). We would ascribe this to either actual cell-to-cell variability concerning the amount of feedback inhibition received, or the number of interneuronal connections conserved within the slice (or a combination of both factors).

For local activation, the spatial profile of inhibition (Figure 3A-C) shows that a population of ‘local’ interneurons, which supply inhibition to adjacent GCs, must be independently excitable by local GC populations. Furthermore, since global GC activation does not increase inhibition beyond the locally elicited maximum (Figure 2I, right panel), all interneurons supplying local inhibition must be already firing at their maximal rates with only local activation. The smaller amplitude of remotely activated inhibition (see Figure 3B, C) implies that a substantial fraction of the interneurons supplying inhibition to the patched GC remains inactive when a remote GC population is activated. A rough estimate of the overall fraction of interneurons recruited by local stimulation can be generated from Figure 3D-E as the average IPSC across all stimulus locations (around ~60% at saturated stimulation powers).

The authors state (subsection “Quantitative physiological properties of DG feedback inhibition”) that the dynamic range is determined by the cellular connectivity patterns, but it is not clear how connectivity could generate this narrow dynamic range – please explain.

The dynamic range of the feedback inhibitory microcircuit will be determined by i) synaptic weights and ii) the divergence/convergence of connections at all participating synapses as well as iii) the active and passive properties of all participating cells (their input-output functions). We have attempted to make this clearer in the manuscript (subsection “Quantitative physiological properties of DG feedback inhibition”). For instance, the lower bound of the dynamic range is defined by the number of GCs needed to first elicit interneuronal APs. The upper bound of the dynamic range is reached when all interneurons anatomically integrated into feedback inhibitory circuits no longer increase their firing rates upon further increased GC activity. In the most parsimonious view, this would be simply because convergent GC synapses are already driving the interneurons to their maximal instantaneous rates. Additionally considering the complex hilar circuitry (mossy cells, disinhibitory connections), the upper bound could also be conceived of as a kind of dynamic equilibrium arising from all participating elements, where for instance maximal inhibition is dampened by disinhibitory interneurons to a degree proportional to the increasing excitation (here dynamic refers not to time, but to active cell fraction). Given these potential complexities (and arbitrarily complex additional circuits can be conjectured), we deliberately focused on the resulting net inhibition of GCs. Regardless of how saturation emerges, it will always be a function of the convergence/divergence of synaptic inputs, the synaptic weights and the input-output functions of participating cells. Also note that, given the high number and dense packing of GCs, 0 to 4% is in fact not all that narrow: Assuming for instance 20000 to 30000 cells in the 350µm slice, 4% would correspond to 800 to 1200 GC.

In measuring feedback IPSCs, the authors addressed voltage escape for dendritic IPSCs (Figure 2—figure supplement 2), but they did not address whether the ChR2-mediated conductance evoked by global light activation affects the measurement of the global max IPSC. In addition, why do the normalized curves not reach 100% (Figure 2 D and H)? Same question for all other figures where "IPSC [%]" is reported.

We were also concerned about the possibility of ChR2 mediated shunting of inhibitory currents, which we cannot categorically exclude. However, the maximal IPSC amplitudes were not different between the electrical and optogenetic stimulations (Figure 2F), which would be expected if there were significant effect of a ChR2 mediated shunting conductance.

The normalized IPSC curves never reach 100% because for each cell IPSCs were normalized to that cells own maximum single IPSC (as stated in the subsection “Electrophysiological Recordings”). This was done throughout the entire manuscript to avoid defining ‘maximum’ based on other properties such as stimulation paradigm, stimulation power, location, etc. Following this normalization IPSCs were averaged over the condition of interest (e.g. stimulation power, stimulation location). Accordingly, all reported normalized IPSCs must by definition remain below 100% (as stated in the aforementioned subsection).

2) Figure 3. The slow rise time of IPSCs seem inconsistent with PV interneurons that are thought to be the major source of feedback inhibition. Presumably the IPSCs are slowed by asynchrony/compound events which might be assessed by quantifying individual events rather than averages to infer PV involvement. Were any PV interneurons recruited in Figure 4? It seems important to address the contribution of PVs since they are critical interneuron subtype in the model. Based on reported uIPSC from PVs, can the authors estimate how many PVs might be contributing to feedback inhibition?

The reviewer raises an interesting question. Unfortunately, our data do not allow us to make valid inferences about the contribution of different interneuron types, since even un-averaged GC IPSCs were compound events. As the reviewers surmise, this asynchrony likely caused the events to be slow. We did not examine in detail which interneurons are responsible for the properties of feedback inhibition, instead concentrating on a quantitative assessment of the input output properties of this circuit. In this sense, we consider the fact that our recorded IPSCs represented non-synchronous integrated events, reflecting the compound activity of various interneuron types biologically arising from GC activity, as an advantage of this experimental approach.

A number of previous studies have, however, addressed the behavior and roles of individual cell-types (summarized in the manuscript in Supplementary file 1). Most pertinent to the present question is perhaps a study by (Stefanelli et al., 2016), who performed exactly the kinetic comparison suggested by the reviewers, and conclude that SST-cells play the most important role in isolated feedback inhibition (through CaMKII-mediated activation of GCs). By contrast, less direct evidence from (Espinoza et al., 2018 and Lee et al., 2016) suggest that PV cells likely also play a role, potentially due to concomitant perforant path input and along with additional interneuron types. However, the optogenetic circuit interrogation by Lee et al. suggests, that together PV and SST- cells still only account for ~50% of inhibition (Figure 6, therein). This may be partially due to the expression patterns of the PV-, SST- and GAD65- mouse lines used, as is usually the case for such transgenic mouse lines.

We are not sure if the interneurons in Figure 4 contain PV interneurons, as we did no post-hoc stainings and used no selective Cre mouse line. However, the recorded cells were all clearly within the hilus (not on the GC border), and had maximal firing rates generally below 60Hz suggesting they did not represent PV^+^ basket cells.

We do not believe the model inferences are critically dependent on the contribution of PV cells, since the experimentally matched properties concerned the spatio-temoral distribution of the arising net inhibition. Which interneurons are the source of this inhibition is, in our view, somewhat incidental (we simply chose a plausible yet parsimonious implementation with just 2 interneuron types for the model).

Accordingly, the two modeled interneuron types could just as well be called local and global interneurons, where each would represent diverse population (including Axo-axonic cells, CCK-BCs, PV-BCs for local; TML and HIPP cells for global).

3) The small number of active PP afferents used (24 of 400) in the input patterns meant that only about 10% of GCs were active even in the absence of inhibition (Figure 6—figure supplement 2). Just as the authors tested the robustness of their conclusions in the face of stronger feed-forward inhibition, it also seems important to assess the robustness of the conclusions across a range of excitatory drive that might represent EC activity under various conditions.

We now include an additional sensitivity analysis concerning PP-drive (new Figure 6—figure supplement 7,subsection “Frequency dependent pattern separation is robust over analysis scales and input strengths”, fourth paragraph). Specifically, we have modified the PP-input weights from 0.6x to 2x leading to GC activity levels of up to 40% in the absence of inhibition. This analysis showed that the frequency dependence was robust over a range of PP-drives (0.8x to 1.6x for 0<R_in_<1, 0.8x to 2x for 0.9< R_in_<1).

4) Subsection “Quantitative physiological properties of DG feedback inhibition”. The authors report that the range of active GCs that saturate feedback inhibition is in the range of active GCs reported in vivo. However, Pernia-Andrade and Jonas, 2014, Pilz et al., 2016, and Diamantaki et al., 2016, report higher fractions of active GCs in vivo, although reports using cFos labeling typically support the 1-4% range. The authors should discuss more explicitly the literature about GC activity in vivo and the interpretations of their small dynamic range of feedback inhibition in light of the possibility that GC activity might not be as sparse as suggested by cFos.

The reviewer points out the important distinction between active cell-fractions that have been reported in the literature. This point is well taken and relies, in our opinion, on the time window in which activity is taken into account. In the electrophysiological experiments cited by the reviewers, cells were defined as active if action potentials were observed in time intervals of >15 minutes (Pernía-Andrade and Jonas, 2014) or > 60 seconds (Diamantaki et al., 2016). However, GCs firing seconds apart will cooperate neither in recruiting inhibition nor in activating CA3-cells. To recruit feedback inhibition, granule cell populations must be concurrently active within a short time window. An electrophysiologically determined firing rate over longer periods can be converted to an active cell fraction within shorter time-windows. If we recalculate the fraction of active GCs in this manner for short time windows of 20 ms (i.e. a gamma cycle) from Pernia Andrade and Jonas, 2014, and Diamantaki et al., 2016, the fraction of active GCs would be 0.4 and 0.028%, respectively (assuming random Poisson firing at the reported frequencies of all active GCs).

Studies relying on Ca^2+^-imaging such as (Allegra et al., 2019; Hainmueller and Bartos, 2018; Pilz et al., 2016) typically report even lower rates with less than 1 transient per minute, though they generally acknowledge that the measured activity may represent only GC bursts.

Finally, the immediate early gene studies report activity integrated over minute long sessions of animal behavior, but likely only capture a small subpopulation of GCs undergoing certain types of plasticity (which likely depend on the specific IEG under investigation).

Accordingly, we believe that the dynamic range we find for coactive cells within ~20ms time-windows is consistent with the literature. In fact, our reading of the rate estimates using these different approaches would suggest that physiological GC activity rarely exceeds 1 or 2% for theta or gamma windows.

We have attempted to incorporate a more explicit, but still brief discussion of this into the manuscript (subsection “Quantitative physiological properties of DG feedback inhibition”). If the reviewer thinks this should be discussed in more detail, we could dedicate a section of the Discussion to the various sources of over- and underestimation errors for the various techniques mentioned above.

5) The nature of the facilitating inhibition remains unclear. Fast-spiking basket cells in the hippocampus usually show paired pulse or multiple pulse depression and for only few interneuron types paired pulse or multiple pulse facilitation has been observed. Whole-cell recordings from GCs during optogenetic stimulation of the different types of DG-interneurons might help to dissect the nature of the facilitating inhibitory inputs (e.g. Somatostatin-expressing cells).

We are not sure if we have conveyed the point we wanted to make well. Unequivocal evidence from dual patch-clamp recordings, which we compiled in Supplementary file 1), has demonstrated that the inhibitory synapses from various interneuron types (including CCK, PV and SST expressing cells) to GCs are depressing (subsection “Short term dynamics in the feedback inhibitory microcircuit”, last paragraph). We have shown that inhibition in the – at least disynaptic – hilar feedback circuit facilitates. We were puzzled by this phenomenon and reasoned that there has to be a facilitating excitatory synapse driving feedback inhibition. Astonishingly, we could not find direct evidence for this in the literature (see Supplementary file 1). Indeed, we found that the MF output that drives feedback interneurons strongly facilitates. We have tried to make this clearer in the revised manuscript (see the aforementioned paragraph).

6) GCs also contact Mossy cells (MCs), which in turn recruit DG interneurons and thereby inhibit GCs. It remained unclear why the MC feedback excitation of hilar interneurons was removed from the network model. It is an important functional element of this circuitry, which provides inhibition to GCs.

We agree with the reviewer, that mossy cells are an important functional element of this circuitry. This seems to be a misunderstanding. We did not remove the MC feedback excitation of hilar interneurons from the network, but the direct MC excitation of GCs. This was done, because within a hippocampal lamella this connection has been shown to be extremely weak (Buckmaster et al., 1996). For instance, Scharfman, 1995, found MC to GC excitation only in 20 out of 1316 cell pairs and only when inhibition was fully blocked.

7) Subsection “Input-output relation of the feedback inhibitory microcircuit”, second paragraph and Figure 1—figure supplement 1 legend – Is it correct that the detection threshold used (0.94%) leads to a "true positive rate of 3%"? That seems very low, and implies that 97% of true responses were not detected. Unless this is a typo, is this not a serious problem that implies that the estimates of the active fraction of GCs are extreme underestimates?

Thank you for this comment. We have corrected this misleading wording. The detection threshold would lead to equal numbers of false positives and false negatives if the ‘actually active fraction of GCs is 3%’. This implies that in this case, the number of false positives and false negatives would cancel each other out.

8) Frequency-dependence of facilitation – Please state explicitly whether there was a "frequency tuning" (e.g., a preferred frequency) or whether all frequencies {greater than or equal to} 10 Hz displayed the same facilitation ratio (greater than 1).

The facilitation indices significantly increased with increasing stimulation frequency (the ANOVA posttest for linear trend was significant (p<0.0001, R²=0.436; 1Hz: 0.99 ± 0.07; 10Hz: 1.41 ± 0.11; 30Hz: 1.83 ± 0.16; 50Hz: 2.09 ± 0.19). We have included this in the manuscript (subsection “Short term dynamics in the feedback inhibitory microcircuit”, first paragraph).

9) Spike rates – Here, pattern separation was computed from Pearson's correlation, dot product and pattern overlap of population spike rate vectors, all of which are in general sensitive to absolute spike rates. Therefore, it is not surprising that some of the manipulations in the model (e.g., reducing various sources of inhibition) would enhance correlations by increasing GC spike rates, thus reducing pattern separation. It would be extremely useful to know if the different outcomes in pattern separation are driven mainly by the impact of the various sources of inhibition on GC spike rate alone, or whether there is indeed something else "special" about the different sources of inhibition, such as their differential IPSC latencies following PP or GC spikes, their IPSC timing jitter or failure rate or their input location onto the GCs. Plots of average GC spike rate versus the area under the ∆R_out_ curve, for the seven model families chosen, would be a first-pass at addressing these questions.

Thank you for this valuable comment. Indeed, our modeling data clearly shows that more is going on than pattern separation through changes in total inhibition and we have not made this sufficiently clear. Firstly, the reviewer is absolutely right in assuming that the main differences in PS when *removing* various sources of inhibition are driven by changes in GC spike rates, which we showed in Figure 6—figure supplement 1C, D). These data confirm the standard account, whereby inhibitory circuits support pattern separation simply by increasing GC sparsity. We have attempted to make this more clear in the manuscript (subsection “Quantitative properties of the feedback circuit predict frequency dependent pattern separation”, third paragraph).

The more interesting and novel finding is that this effect is frequency dependent. Namely, given the exact same mean PP input drive, and the exact same inhibitory circuit, spike modulation at 30Hz leads to significantly lower GC-spiking and better pattern-separation (Figure 6E). Remarkably, this decreased GC spiking is achieved with lower interneuron activity rates in 30Hz vs. 10Hz for both BCs and HCs. We now included a new supplementary figure showing the mean active cell fractions and firing rates for each cell type (new Figure 6—figure supplement 1C, D). Accordingly, the 30Hz modulation more efficiently converts interneuron spiking to GC depression than the 10Hz modulation. Our analysis (Figure 6E) showed that this frequency dependent effect is mediated by the feedback circuit but not the feedforward circuit, implying that it is not simply the amount of inhibition but rather the timing, as the reviewer suspected.

Furthermore, the most prominent frequency-dependent boosting of feedback inhibitory (but not feedforward inhibitory) pattern separation, is seen for the patterns with the highest similarity (Figure 6E, only R_in_>0.9; Figure 6G, arrows; subsection “Quantitative properties of the feedback circuit predict frequency dependent pattern separation”, fourth paragraph). Here GC sparsity is by definition constant, since the R-values are computed on the exact same model runs.

Based on the reviewer’s suggestion, we now include an additional analysis relating pattern separation to GC spiking directly for a range of PP input weights (new Figure 6—figure supplement 7). Firstly, these data demonstrate that the frequency dependent effect is robust over the range of PP-drives leading to plausible GC activity levels (Figure 6—figure supplement 7A). They also suggest that the degree of pattern-separation observed is mostly a function of the achieved GC sparsity when effects are investigated over all input similarities (Figure 6—figure supplement 7E)30Hz data are consistently shifted towards more sparsity and better pattern separation, but they appear to be roughly on the same line. I.e. decreasing PP-drive until a network at 10Hz shows the same sparsity as it would at 30Hz leads to the same approximate degree of pattern separation. However, for highly similar input patterns (Figure 6—figure supplement 7D, E), this was not the case, with 30Hz data showing better pattern separation even if PP-drive is reduced to achieve the same sparsity (subsection “Frequency dependent pattern separation is robust over analysis scales and input strengths”, fourth paragraph).

All this clearly demonstrates that more is going on than a simple regulation of sparsity through changes in total inhibition. We have attempted to make this clearer in the manuscript. We are not sure which seven model families the reviewer is referring to. There are three basic model families with associated spike rates (full model, no feedback, no inhibition). The ‘no feedback’ and ‘no inhibition’ data are computed from these three models (e.g. ‘no feedback’ shows the difference between ‘full model’ and ‘no feedback’). The other two model families (no facilitation, no spatial tuning) seem of limited interest here, since they showed no major effects.

10) "The effect of facilitation on pattern separation is intuitive, since this allows the feedback circuit to integrate GC activity over time, and convert it to inhibition." Reviewers were not sure how intuitive this really is. True, integrating GC activity over time might be useful, but a) depression would allow a mathematical differentiation (or high-pass filtering) of GC activity over time, which arguably could be better than integrating over time for the purpose of separating temporal patterns, and b) the output of the inhibitory circuit is largely depressing anyway, providing the differentiation mentioned above. Thus I think the question remains: what is the purpose of the facilitation of the GC->BC synapse if the ultimate output will be depression at the BC->GC synapse anyway. Indeed, the authors found that the GC->BC facilitation only had a small effect on pattern separation.

This is an interesting point. It is true that, depending on the definition of pattern separation (temporal pattern separation between spike trains vs. spatial pattern separation of coactive GC ensembles) the effects of facilitation would differ. Given that we did observe pronounced facilitation we can only speculate as to its purpose. We do believe that facilitation may aid sparse GC spiking to recruit inhibition. Indeed, the view that a subsequent depressing synapse negates a previous facilitating one is problematic in sparsely firing networks, since the facilitation may be necessary to bring interneurons to threshold in the first place. The reviewer is however correct in noting that the contribution of facilitation to pattern separation was relatively small in our simulations. We have changed the Discussion to a more cautious interpretation (subsection “Spatiotemporal organization of inhibition and pattern separation”, first paragraph).